# Combination treatment of berberine and solid lipid curcumin particles increased cell death and inhibited PI3K/Akt/mTOR pathway of human cultured glioblastoma cells more effectively than did individual treatments

**Panchanan Maiti**[1,2,3,4,5]*, **Alexandra Plemmons**[1], **Gary L. Dunbar**[1,2,3,4]*

**1** Field Neurosciences Institute Laboratory for Restorative Neurology, Central Michigan University, Mt. Pleasant, MI, United States of America, **2** Program in Neuroscience, Central Michigan University, Mt. Pleasant, MI, United States of America, **3** Department of Psychology, Central Michigan University, Mt. Pleasant, MI, United States of America, **4** Field Neurosciences Institute, Ascension of St. Mary's Hospital, Saginaw, MI, United States of America, **5** Department of Biology, Saginaw Valley State University, Saginaw, MI, United States of America

* maiti1p@cmich.edu(PM); dunba1g@cmich.edu(GLD)

## Abstract

The treatment of glioblastoma is challenging for the clinician, due to its chemotherapeutic resistance. Recent findings suggest that targeting glioblastoma using anti-cancer natural polyphenols is a promising strategy. In this context, curcumin and berberine have been shown to have potent anti-cancer and anti-inflammatory effects against several malignancies. Due to the poor solubility and limited bioavailability, these compounds have limited efficacy for treating cancer. However, use of a formulation of curcumin with higher bioavailability or combining it with berberine as a co-treatment may be proving to be more efficacious against cancer. Recently, we demonstrated that solid lipid curcumin particles (SLCPs) provided more bioavailability and anti-cancer effects in cultured glioblastoma cells than did natural curcumin. Interestingly, a combination of curcumin and berberine has proven to be more effective in inhibiting growth and proliferation of cancer in the liver, breast, lung, bone and blood. However, the effect of combining these drugs for treating glioblastoma, especially with respect to its effect on activating the PI3K/Akt/mTOR pathways has not been studied. Therefore, we decided to assess the co-treatment effects of these drugs on two different glioblastoma cell lines (U-87MG and U-251MG) and neuroblastoma cell lines (SH-SY5Y) derived from human tissue. In this study, we compared single and combination (1:5) treatment of SLCP (20 μM) and berberine (100 μM) on measures of cell viability, cell death markers, levels of c-Myc and p53, along with biomarkers of the PI3K/Akt/mTOR pathways after 24–48 h of incubation. We found that co-treatment of SLCP and berberine produced more glioblastoma cell death, more DNA fragmentation, and significantly decreased ATP levels and reduced mitochondrial membrane potential than did single treatments in both glioblastoma cells lines. In addition, we observed that co-treatment inhibited the PI3K/Akt/mTOR pathway more efficiently than their single treatments. Our study suggests that

**Data Availability Statement:** All relevant data are within the paper and its Supporting Information files.

**Funding:** This research work was supported by Field Neurosciences Institute of Ascension Health and generous donations from Clif and Diane Rafter, Stan Fawcett, and Robert Schellhas, the Neuroscience Program and the John G. Kulhavi Professorship at Central Michigan University. We thank Verdure Science (Noblesville, IN) for donating Solid lipid curcumin particle for this study.

**Competing interests:** The authors have declared that no competing interests exist.

**Abbreviations:** GB, glioblastoma multiforme; Cur, curcumin; SLCP, solid lipid curcumin particles; BBR, Berberine; WHO, World Health Organization; TMZ, temozolomide; ROS, reactive oxygen species; NF-kB, nuclear factor kappa beta; Jak1, janus kinase-1; STAT3, signal transducer and activator of transcription 3; mTOR, mammalian target of rapamycin; VEGFR2, vascular endothelial growth factor receptor 2; ERK, extracellular receptor kinase; PI3K, phosphoinositol triphosphate; MTT, 3-(4,5-dimethylthiazol-2-yl)-2,5-diphenyltetrazolium bromide; PI, propidium iodide; EtBr, ethidium bromide; TUNEL, terminal deoxyribonucleic acid nick end labeling; PVDF, polyvinylidene difluoride; HRP, Horseradish peroxidase; DAPI, 4′,6-diamidino-2-phenylindole; EMEM, Eagle's Minimum Essential Medium; DMEM, Dulbecco's modified Eagle's medium; ECACC, European Collection of Authenticated Cell Cultures; ATCC, American type culture collection; FBS, fetal bovine serum; IHC, immunohistochemistry; HBSS, Hank's balanced salt solution; BrdU, bromodeoxyuridine; AU, arbitrary unit; SEM, standard error of mean; FITC, fluorescent isothiocyanate; SCGE, Single cell gel electrophoresis; DCFH, DA-dichloro-dihydro fluorescein diacetate; RIPA, radio immunoprecipitation assay; OD, Optical density; GAPDH, glyceraldehyde phosphate dehydrogenase; ANOVA, One way analysis of variance; HSD, honestly significant difference; MMP, mitochondrial membrane potential; Bax, Bcl2-associated X protein; Cyt-c, Cytochrome-c; Bcl2, B-cell lymphoma 2; FDA, Food and Drug Administration; EGFR, extracellular growth factor receptor; LDH, lactate dehydrogenase; AIF, apoptotic inducing factor; GluT1, glucose transporter 1; HK-2, hexokinase-2; MAPK, mitogen activated protein kinase.

combination treatments of SLCP and berberine may be a promising strategy to reduce or prevent glioblastoma growth in comparison to individual treatments using either compound.

# 1. Introduction

Glioblastoma (GB), or grade-IV astrocytoma, is one of the most aggressive and deadliest brain cancers, killing millions of people world-wide [1]. Patients often cannot survive more than 15–20 months following an initial diagnosis [1]. Current treatment strategies, including surgical removal of tumor, radiotherapy, and chemotherapies, or combinations of these therapies are unable to stop the progression of this disease. In this context, temozolomide (TMZ), a DNA-alkylating chemotherapeutic agent, has been used to treat GB for more than two decades. Although it does provide some modest survival benefits for GB patients [2–4], its propensity to trigger immunoresistance and neuroinflammation [5, 6] make it less attractive for treating GB patients. Therefore, alternative therapies that reduce neuroinflammation, as well as prevent or slow GB invasion and metastasis are necessary. One approach involves the use of natural poly-phenols, such as curcumin (Cur) and berberine (BBR), which exhibit antiproliferation and anti-cancer properties in human malignancies. These compounds are being tested for their anti-cancer effects on a variety of cancer types, including GB [7–11]. Curcumin (Cur) is a yellow colored phytopolylphenol derived from the root of the herb *Curcuma longa* [12]. It inhibits tumor growth by suppressing cellular transformation, proliferation, invasion, angiogenesis, and metastasis [11, 13, 14]. Higher concentrations of Cur induces apoptosis in cancer cells [9, 15, 16] by increasing reactive oxygen species (ROS), inhibiting the PI3K/AKT/mTOR pathway and inhibiting NF-kB signaling in human neuroblastoma [8]. Cur also attenuates glioma growth by inhibiting the JAK1,2/STAT3 signaling pathway in a syngeneic mouse model [17]. Unfortunately, natural Cur has limited solubility and is unstable in physiological fluids, due to its hydrophobic and lipophilic nature, limiting its bio-availability. Recently, we and others have demonstrated that lipid-conjugation increases Cur solubility and bio-availability in cancer therapy [16, 18, 19].

Similarly, berberine (BBR), an isoquinoline alkaloid isolated from *Berberis vulgaris L.*, has been used extensively in traditional Chinese medicine to treat diarrhea and diabetes. It exhibits anti-cancer activity in glioma [20], colorectal [21]-, lung [22] -, prostate [23]—and ovarian cancer [24], by inducing apoptotic cell death. Recently, Wang and colleagues reported that BBR induces autophagy in GB by targeting the AMPK/mTOR/ULK1-pathway [25]. In addition, Jin and colleagues reported that BBR inhibits angiogenesis in GB xenografts by targeting the vascular endothelial growth factor receptor-2 (VEGFR-2) and the extrallular receptor kinase (ERK) pathway [26]. Furthermore, Agnarelli and colleague also showed that berberine induced autophagy in U343 GB cells [27].

As the PI3K/Akt/mTOR signaling pathway plays a pivotal role in GB survival, inhibiting this pathway using anti-cancer natural polyphenols offers a viable strategy to prevent GB growth. Recently, we have shown that when using solid lipid curcumin particles (SLCPs), stronger anti-cancer effects and the greater inhibition of the PI3k/Akt/mTOR pathway in human GB cell were observed than when natural Cur was used [28]. Inhibition of GB growth requires a significant amount of free Cur, which is quite difficult to deliver to the cancer cells. Moreover, because of the solubility and bioavailability issues in both Cur and BBR, it is difficult to get sufficient levels of these compounds to induce significant GB cell death after their oral administration. Therefore, we hypothesized that co-treatment of Cur with BBR would enhance their anti-cancer effects.

Recently, Wang and colleagues demonstrated the synergistic chemopreventive effects of Cur and BBR on human breast cancer cells through induction of apoptosis and autophagic cell death [29]. Similarly, Balakrishna and colleagues reported that co-treatment of Cur and BBR induced more cell death than did individual treatments on A549, Hep-G2, MCF-7, Jurkat, and K562 cell lines [30]. These findings provided us with a strong rationale to investigate the co-treatment of these two anti-cancer drugs to prevent GB growth and proliferation.

The present study was designed to compare the combination of Cur and BBR treatments with single treatments of SLCP and BBR in two different GB cell lines derived from human tissue. We have investigated the cell viability, DNA fragmentation, cell death mechanism, mitochondrial dysfunction, ATP levels and the PI3K/Akt/mTOR signaling pathway. Our results suggest that the co-treatment of SLCP and BBR caused more cell death and inhibited the PI3K/Akt/mTOR pathway more efficiently than did either individual treatment.

## 2. Materials and methods

### 2.1. Chemicals

Berberine (catalog no: B3251-10G), MTT [3-(4,5-dimethylthiazol-2-yl)-2,5-diphenyltetrazo-lium], propidium iodide (PI), ethidium bromide (EtBr), agarose, proteinase-K protease inhibitor cocktail (catalog no: P8340-5ML) and other accessory chemicals were procured from Sigma (St. Louis, MO). An in situ BrdU-Red DNA fragmentation assay kit (TUNEL staining kit, catalog no: ab66110) and Annexin-V staining kit (catalog no: ab176749) were purchased from Abcam (Cambridge, MA). Low melting agarose was from Invitrogen (Grand Island, NY; catalog no: 16520050). Cell-ROX® reagent and CyQUANT Cell Proliferation Assays Kits (catalog no: C35006) were from Molecular Probe (Grand Island, NY). Polyvinylidene difluoride (PVDF) membrane and ImmobilonTM Western Chemiluminescent HRP-substrate were from (Millipore, Bedford, MA). Hoechst 33342 trihydrochloride-trihydrate solution was purchased from ThermoFisher Scientific (Grand Island, NY). 4′,6-diamidino-2-phenylindole (DAPI) was from IHC-World (Ellicott City, MD). Tris-glycine gel (4–20%, catalog no: XP04200BOX) was from Invitrogen (Carlsbad, CA). Cell culture media, such as EMEM and DMEM: F12K were procured from the American Type Culture Collection (Manassas, VA). SLCPs, which contain 26% pure Cur, was gifted from Verdure Sciences (Noblesville, IN). These SLCPs consist of high-purity, long-chain phospholipid bilayer and a long-chain fatty acid solid lipid core, which coats the Cur. The SLCPs have been well characterized by us and others in collaboration with Verdure Sciences [31–35], including clinical studies in Alzheimer's disease [36]. The U-87MG (catalog no: ATCC® HTB-14™), SH-SY5Y (catalog no: ATCC® CRL-2266™) and N2a (catalog no: ATCC® CCL-131™) were purchased from American Type Culture Collection (ATCC, Manassas, VA), whereas U-251MG cell line was purchased from European Collection of Authenticated Cell Cultures (ECACC, catalog no: 09063001).

### 2.2. Cell culture

Two GB cell lines (U-87MG and U-251MG) and two cortical neuronal cell lines (SH-SY5Y and N2a) were used in this study. Briefly, the U-87MG, U-251MG and N2a cells were grown with Eagle's Minimum Essential Medium (EMEM) containing 10% heat-inactivated fetal bovine serum (FBS) and penicillin/streptomycin (pen/strep, 1μg/mL), whereas SH-SY5Y cells were grown in Dulbecco modified Eagles' medium and F12K (DMEM:F12K, 1:1) along with 10% FBS and pen/strep, (1μg/mL). The cultures were maintained at 37˚C in a humidified atmosphere at 5% $CO_2$. For cell viability and cell death assays, the cells were grown in 96-well plates, whereas for immunohistochemistry (IHC) cells were grown on glass coverslips, with

fresh EMEM or DMEM: F12K and antibiotics, but without growth factors, depending on the experimental setup. For Western blot techniques, cells were grown on 60-mm Petri plates in EMEM, or DMEM: F12K without growth factors.

## 2.3. Treatment of SLCP and Berberine

Because Cur and or BBR have greater solubility in methanol [28, 37], both were dissolved in pure methanol (100%) and then diluted in Hank's balanced salt solution (HBSS) to obtain their desired concentration before being added to the 96-well plate, glass cover slips or Petri-dishes containing the cells. The final methanol concentration was <0.1%. After performing a dose response study, the final concentrations chosen were 20 μM for SLCP and 100 μM for BBR, whereas combination treatment of SLCP and BBR was 1:5 (20 μM +100 μM). All the cell lines were treated with these two drugs for 24–48 h for cell viability, cell death and for immunocytochemistry assays, whereas for Western blots, cells were treated for 24 h.

## 2.4. Cell viability by MTT assay

A cell viability test was performed using an MTT [3-(4,5-dimethylthiazol-2-yl)-2,5-diphenyl-tetrazolium bromide] assay, as described previously [35, 38, 39]. The cells were treated with different concentrations of SLCP (in μM: 5, 10, 20 and 40) and BBR (in μM: 50, 100, 150 and 200) for 24 h. After standardization of toxicity levels, 20 μM of SLCP, and 100 μM of BBR and their combination (1:5) were used for all experiments with 24-h exposure. Based on the cell viability data, we observed that in the case of SLCP, after the concentration of SLCP was increased above 20 μM, we observed cell death. Similarly, for berberine, concentrations above 100 μM did not result in further cell death. Therefore, we decided to use 20 μM of SLCP and 100 μM of berberine (1:5). The optical density was measured at 570 nm using a Synergy plate reader (Bio-TEK instruments, Winooski, VT). The results of the five independent experiments (6 wells per condition) were normalized to the medium control group and expressed as mean ± SEM.

## 2.5. Cell Proliferation assay

Cell proliferation was assayed using a CyQUANT® NF Cell Proliferation Assay Kit to provide an accurate and simple measure of cell number. The assay is based on the incorporation of thymidine analogs, such as 3H thymidine or bromodeoxyuridine (BrdU) during DNA synthesis, or on measurement of metabolic activity indices, such as oxidoreductase activity or ATP levels. It measures the cellular DNA content via fluorescent dye binding, which is closely proportional to cell number. The protocol was followed as per the manufacturer's instruction. Briefly, after plating the cells overnight in EMEM, treatments with SLCP, BBR or their combination were applied 24 h. On the next day, the growth medium was gentley aspirated, and 100 μL of 1X dye binding solution [22 μL of CyQUANT NF dye reagent (Component A) was added to 11 mL of 1X HBSS buffer], which, in turn, was added to each well and allowed to incubate for 60 minutes at 37˚C in the dark. The fluorescence intensity was measured from each sample using a fluorescence microplate reader (Bio-TEK instruments, Winooski, VT) with excitation at ~485 nm and emission detection at ~530 nm. The mean fluorescent intensity (arbitrary unit, AU) from the three independent experiments (6 wells per condition) were normalized to the medium control group and expressed as mean ± SEM. In addition, cells were also imaged using a fluorescent microscope (Leica, Germany) with appropriate filters.

## 2.6. DNA fragmentation study by TUNEL staining

DNA fragmentation was performed using an *In Situ* BrdU-Red DNA Fragmentation (TUNEL) Assay Kit, as per manufacturer instructions [35, 38]. Briefly, U-87MG and U-251MG cells were grown on coverslips overnight in EMEM, without any growth factors, and were treated then with SLCP (20 μM), BBR (100 μM) or their combination (1-part SLCP to 5 parts BBR) for 48 h. Following treatment, the cells were fixed with 4% paraformaldehyde for 15 min, and then TUNEL staining was performed, as described previously [35, 38]. Finally, the cells were counter-stained with Hoechst 3342 or DAPI for 10 min at room temperature. Images were taken using a fluorescent microscope (Leica, Germany), with appropriate filters (excitation/emission: 488/576). The red fluorescent signal indicated TUNEL-positive cells. The number of total cells and TUNEL-positive cells were counted by two individual researchers and expressed as a percentage of TUNEL-positive cells. More than fifty microscopic fields were randomly selected for counting the number of TUNEL-positive cells from two independent experimental setups and these were used to obtain a mean value.

## 2.7. Annexin-V staining for apoptotic cell death

The Annexin-V staining was performed, as described previously [28, 35, 40]. Briefly, the U-87MG cells were treated with SLCP (20 μM), BBR (100 μM) or their combination (using this 1:5 ratio) for 24 h and then annexin-V-FITC staining was performed, along with counter-staining with Hoescst-3224 (1μg/ml) [35]. The total number of cells and the number of annexin-V-positive cells (green) were counted per microscopic field and expressed as a percentage of dead cells. Approximately 30 microscopic fields (~5000 total cells) from two independent experimental setups were used for counting.

## 2.8. Single-cell gel electrophoresis (SCGE) or comet assay

The comet assay was performed to measure the degree of DNA strand breaks, as described previously [41–43]. The detail protocol for SCGE was described by us previously [28].

## 2.9. JC-1 stain and confocal imaging

JC-1, a membrane permeable fluorescent dye which is widely used for monitoring mitochondrial health and cell death. It is considered as a good indicator of mitochondrial membrane potential (MMP) in neurons, as well as in intact tissues and isolated mitochondria. This dye accumulates in mitochondria with potential-dependent, which can be monitored by flow cytometry or by fluorescent microscopic imaging. JC-1 staining protocol was followed as per manufacture instruction. Briefly, U-87MG and U-251MG were grown overnight on poly-D-lysine coated glass cover slips in EMEM (1x10$^5$/ml) without growth factors. On the next day, the cells were treated with SLCP, BBR, and their combination (1-part SLCP to 5 parts BBR). After 24 h of the drug treatment, the media was discarded, the cells were washed with Dulbecco's phosphate buffer saline (DPBS) and incubated with JC-1 dye (dissolved in DMSO, to a final concentration of 2 μM) at 37°C, in 5% $CO_2$, for 15 to 30 minutes. The cells were washed in warm DPBS three times and then fixed with 4% paraformaldehyde solution for 10 min. After fixation, the cells were washed with PBS two times, followed by counter-staining with DAPI for 10 min at room temperature on a shaker in the dark. The cells were washed with distileed water and dehydrated, mounted, and visualized using a confocal laser scanning microscope with a 60x objective at three times optical zoom (total magnification: 1800x) using appropriate excitation/emission filters. Fifteen to twenty randomly selected microscopic images were randomly selected from each group of samples from three independent

experiments and the number of clearly visible mitochondria (red dots) were counted manually from 10–15 cells in each group and expressed as mean ± SEM.

## 2.10. Detection of reactive oxygen species (ROS)

Intracellular accumulation of ROS was detected by 2'-7'-dichlorodihydrofluorescein diacetate (DCFH-DA), using a CellRox assay, as described previously [7, 28, 35, 44]. The presence of green fluorescent signal indicated ROS levels and use of CellROX dyes provided a conventional probe for measuring oxidative stress. Total fluorescent intensity (AU) of individual cells was measured using Image-J software (https://imagej.nih.gov/ij/), and at least 200–300 cells were randomly selected from two independent experiments to obtain a mean value.

## 2.11. Immunocytochemistry

Immunocytochemistry of anti-caspase-3, p53, and c-Myc were performed as described previously [28, 35].

## 2.12. DNA gel electrophoresis

DNA gel electrophoresis was performed to measure the DNA fragmentation, as described previously [28, 45].

## 2.13. Western blot

To investigate the different protein markers related to cell survival, cell death and other signaling pathways, Western blots were performed as described previously [28, 35]. Briefly, after the stipulated period of time for each experiment, the U-87MG and U-251MG cells were lysed with cold radioimmuno-precipitation assay (RIPA) buffer, along with protease inhibitor cocktail (Sigma). Equal amounts of protein were loaded in each lane and electrophoresed on 4–20% Tris-glycine gel and transferred to PVDF membrane. After probing with respective primary (**Table 1**) and secondary antibodies, the blots were developed with Immobilon™ Western Chemiluminescent HRP-substrate. The relative optical density (OD) was measured using Image-J software (https://imagej.nih.gov/ij/). To ensure equal protein loading in each lane, the blots were re-probed with either β-tubulin or GAPDH. The values were expressed as mean ± SEM from at least two independent experiments.

## 2.14. Statistical analysis

The measures for cell viability, cell death, immunofluorescent intensity, Western blots and other parameters were expressed as a mean ± SEM. The data were analyzed using one-way analysis of variance (ANOVA), followed by *post-hoc* Tukey HSD (honestly significant difference) tests (https://astatsa.com/OneWay_Anova_with_TukeyHSD). Probability ≤0.05 was considered as statistically significant.

# 3. Results

## 3.1. Combination treatment with SLCPs and BBR reduced cell viability more in U-87MG and U-251MG cells than either SLCP or BBR alone

To compare the cell death caused by SLCP + BBR and their single treatment, we performed an MTT reduction assay with different doses of SLCP (5-, 10-, 20- and 40- μM) and BBR (50-, 100-, 150- and 200- μM) and their combination (at the 1:5 ratio) for 24 h. We found that SLCP +BBR induced ~50% cell death, whereas SLCP induced ~32% cell death and BBR induced

**Table 1. Sources of different antibodies used in this study.**

| Antibodies | Source | Type | Company | Catalog no. | Address |
|---|---|---|---|---|---|
| Bax | Rabbit | Polyclonal | Cell signaling Technology | 2772S | Danvers, MA |
| Bcl-2 | Mouse | Monoclonal | Santa Cruz Biotech | Sc-7382 | Santa Cruz, CA |
| Cyt-c | Rabbit | Monoclonal | Cell Signaling Technology | 4272S | Danvers, MA |
| Caspase-3 | Rabbit | Monoclonal | Cell Signaling Technology | 9661S | Danvers, MA |
| p53 | Rabbit | Polyclonal | Cell signaling Technology | 9282 | Danvers, MA |
| c-Myc | Rabbit | Polyclonal | Cell signaling Technology | 9402 | Danvers, MA |
| Akt | Rabbit | Monoclonal | Cell signaling Technology | 9272S | Danvers, MA |
| pAkt (Ser473) | Rabbit | Monoclonal | Cell signaling Technology | 4060S | Danvers, MA |
| PI3Kp85 | Rabbit | Polyclonal | Cell signaling Technology | 4292S | Danvers, MA |
| p-PI3Kp85 (Tyr458)/p55 (Tyr199 | Rabbit | Polyclonal | Cell signaling Technology | 4228S | Danvers, MA |
| mTOR | Rabbit | Polyclonal | Cell signaling Technology | 2972S | Danvers, MA |
| p-mTOR | Rabbit | Monoclonal | Cell signaling Technology | 2971S | Danvers, MA |
| GAPDH | Rabbit | Monoclonal | Cell signaling Technology | 2118S | Danvers, MA |
| β-tubulin | Rabbit | Monoclonal | Cell signaling Technology | 2146S | Danvers, MA |

~40% cell death after 24h in U-87MG cells (Fig 1A and 1B). In the case of U-251MG cells, cell death was 42% in the presence of SLCP+BBR, 25% in the presence of SLCP and 35% in the presence of BBR (Fig 1A and 1C). In contrast, we found ~10–15% reduction of cell viability in the case of N2a cells after co-treatment of SLCP (20 μM) and BBR (100 μM), and 19% cell death in the case of SLCP+BBR-treated SH-SY5Y cells (Fig 1D and 1E). In human neuroblastoma cells (SH-SY5Y), BBR (100 μM) caused a 10% increase in cell death and SLCP+BBR-treated cells showed 38% reduction of cell viability, without evidence of toxicity in the case of SLCP-treated SH-SY5Y cells (Fig 1E).

### 3.2. Combination treatment of SLCP and BBR reduced cell proliferation in U-87MG and U-251MG cell lines more than their individual treatments

Cell proliferation assays were performed to provide an accurate and simple measure of cell number after treatment with different doses of SLCP, BBR or with SLCP+BBR (in a 1:5 ratio). The fluorescent intensity (arbitrary unit; AU) was significantly less in both SLCP-treated U-87MG cells (for 5 μM: 13.42%; 10 μM: 21.68%; 20 μM: 29.84%; 40 μM:32.48%); and BBR-treated U-87MG cells (50 μM: 1.76%; 100 μM:17.81%; 150 μM: 22.91%; and 200 μM: 24.25%) as compared to the vehicle treatment. Furthermore, fluorescent intensity was decreased by 32.47% (p<0.01) in SLCP+BBR (20 μM +100 μM)-treated U-87MG cells compared to the vehicle (Fig 2A and 2B). Similarly, the fluorescent intensity (AU) was significantly reduced in U-251MG cells in the case of SLCP- (for 5 μM: +4.20%; 10 μM: 19.29%; 20 μM: 46.19%; and 40 μM:49.96%) and BBR-(50 μM: 30.02%; 100 μM: 33.12%; 150 μM: 44.37%; and 200 μM: 56.15%) treated cells. Additionally, fluorescent intensity was decreased by 56.01% (p<0.01) in the SLCP+BBR- (20 μM +100 μM) treated U-87MG cells (Fig 2A and 2C).

### 3.3. Combination treatment of SLCP and BBR increased DNA fragmentation in U-87MG and U-251MG cell lines more than individual treatments

In situ BrdU-Red DNA fragmentation assays, also known as terminal deoxyribonucleic acid nick-end labeling (TUNEL), were performed to investigate the number of DNA fragmented

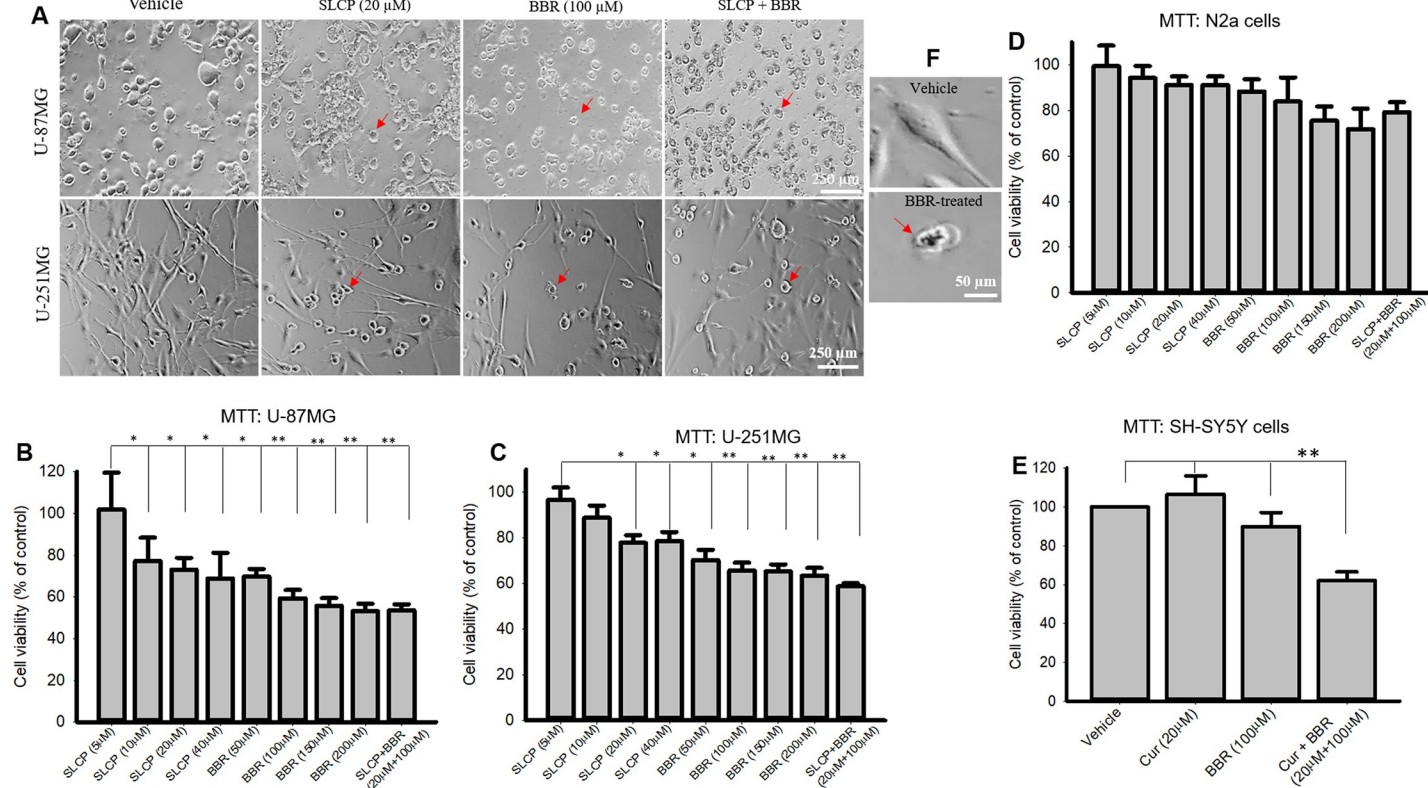

**Fig 1. Morphology and cell viability in U-87MG, U-251MG, SHSY5Y and N2a cells after treatment with SLCP, BBR and SLCP+BBR.** U-87MG and U-251MG cells were grown in EMEM and treated with either SLCP (20 μM), BBR (100 μM) or their combination (in this 1:5 ratio) for 24 h, after which an MTT assay was performed. **A:** An increased number of pyknotic-like cells in both U-87MG and U-251MG cells were observed with SLCP, BBR and the combined (SLCP+BBR) treatment. **B-C:** Cell viability was decreased in both the cell lines in a dose-dependent manner after treatment with SLCP, BBR or SLCP + BBR. Note that the combination treatment showed decreased cell viability comparable to the higher doses of the individual treatments of SLCP and BBR. **D-E:** Cell viability was unaltered in the case of N2a (D) and SH-SY5Y cells after 24-h of SLCP and or BBR, although co-treatment caused a 40% increase in cell death in SH-SY5Y cells (**E**). Arrows indicate pyknotic cells. **F:** showing normal and pyknotic cells in higher magnification. Scale bar indicates 250 μm and is applicable to all images. *p<0.05 and ** p<0.01 in comparison to vehicle-treated cells.

cells after treatment with SLCP, or BBR, or their combination. We found an increased number of TUNEL-positive U-87MG cells in the case of SLCP-treated (22.30%; p<0.01) and BBR-treated cells (26%; p<0.01) in comparison to untreated cells after 24 h. Furthermore, SLCP +BBR-treated cells showed 47% (p<0.001) TUNEL-positive cells in U-87MG cells (Fig 3A and 3C).

Similar trends were observed in U-251MG cells. SLCP treatment showed a 32.45% (p<0.001) increase in TUNEL-positive cells, while BBR treatment caused a 24.70% (p<0.001) increase and SLCP+BBR produced a 38.86% (p<0.001) increase in TUNEL-positive cells after 48 h of treatment (Fig 3B and 3D).

### 3.4. Combination treatments of SLCP and BBR increased the amount of Annexin-V-positive labeling in U-87MG and U-251MG cells more than did individual treatments

We performed Annexin-V staining to quantify apoptotic cell death as a marker of increased efficacy of SLCP+BBR treatments as compared to the individual drug treatments. We observed an increased number of Annexin-V-positive U-87MG cells in the case of SLCP-treated (32.33%; p<0.001) and BBR-treated cells (34.06%; p<0.001) in comparison to untreated cells

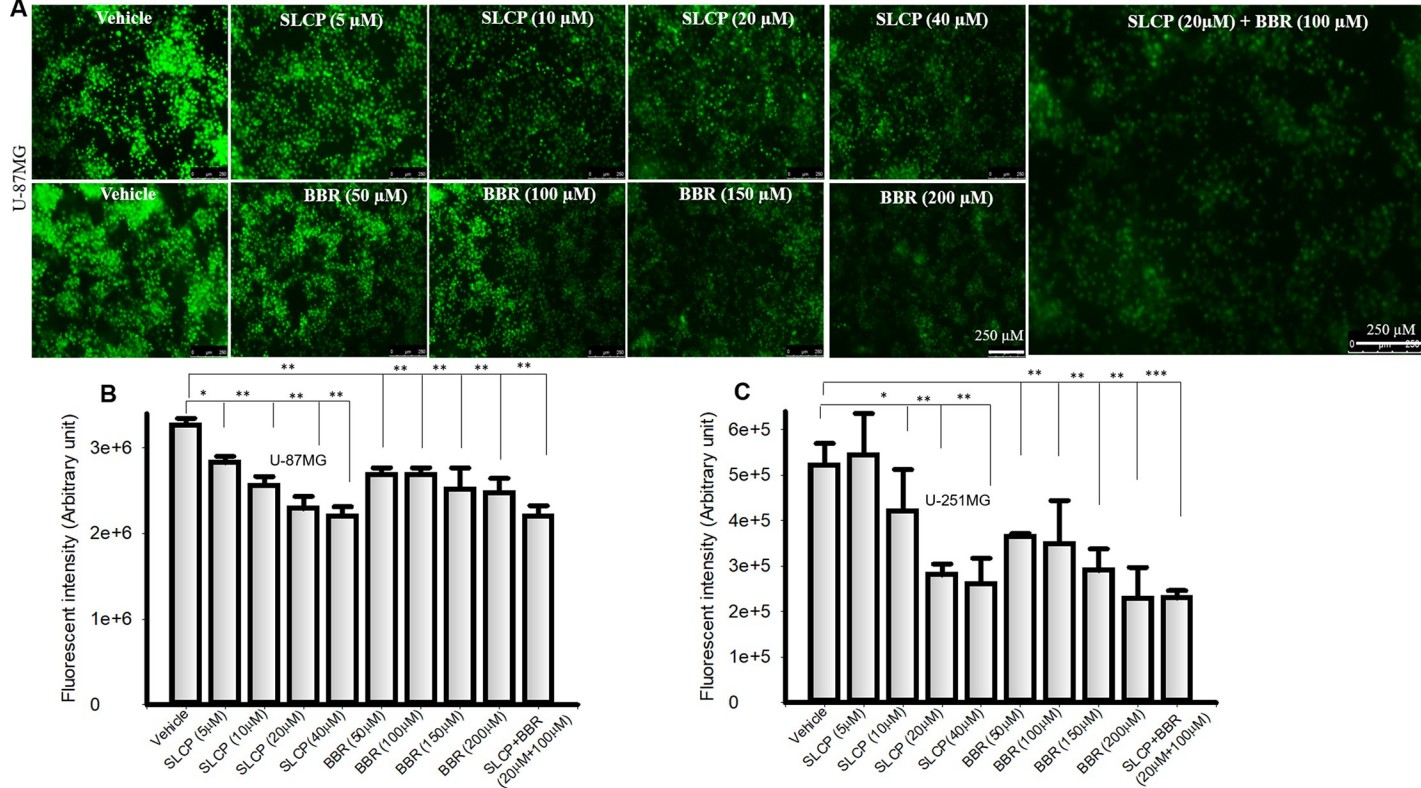

**Fig 2. Cell proliferation was decreased more by combination treatment of SLCP and BBR than individual treatments.** U-87MG and U-251MG cells were grown in EMEM and 1% pen/strep for 24 h and then treated with either SLCP (20 μM), or BBR (100 μM) or their combination for 24 h, after at which a cell proliferation assay was performed. **A**: Immunofluorescent signal was decreased in a dose-depend fashion in U-87Mg cells after reatment with SLCP and BBR. Note that combination treatment (right image) caused a greater decrease in the immunofluorescent signal than did the single treatment (p<0.01). **B-C:** Both U-87MG and U-251MG cells showed a significant decrease in immunofluorescent intensity (arbitrary unit) after treatment with SLCP and BBR. Scale bar indicates 250 μm and is applicable to all images. *p < .05 and ** p<0.01 in comparison to vehicle-treated cells.

after 24 h, whereas SLCP+BBR-treated cells showed a 55.87% (p<0.001) increase in Annexin-V-positive U-87MG cells (**Fig 4A and 4C**). Similarly, in the case of U-251MG cells, SLCP treatments caused a 30.44% (p<0.001) increase in Annexin-V-positive labeling, while BBR produced a 31.06% (p<0.001) increase and SLCP+BBR treatments induced a 57.42% (p<0.001) Annexin-V-positive labeling after 24 h of treatments (**Fig 4A and 4C**). To corelate the cell death, we also performed LDH release assay in U-87MG cells after treatment with SLCP and or BBR or their combination for 48h. We observed that co-treatment of SLCP and BBR increased more cytotoxicity (307.07%) in comparison to SLCP (291.45%) and or BBR (237.29%) alone (**Fig 4D**).

### 3.5. Combination treatment of SLCP and BBR induced more comet-positive cells and increased nuclear lobe formation in U-87MG cells than than did individual treatmkents of either SLCP or BBR

DNA fragmentation is an important phenomenon observed in cell death and the comet assay is the gold standard method for quantifying the degree of DNA fragmentation *in vitro*. Significantly more comet-positive cells were observed in SLCP+BBR-treated cells after 24- and 48-h than in SLCP- or BBR-treated cells (SLCP: 34.00%; BBR: 37.88%; SLCP+BBR: 51.04%; p<0.001) (**Fig 5A–5C**). All three treatments had more comet-positive cells in comparison to the vehicle-treated control U-87-MG cells (**Fig 5A–5C**). A similar trend was observed after 48

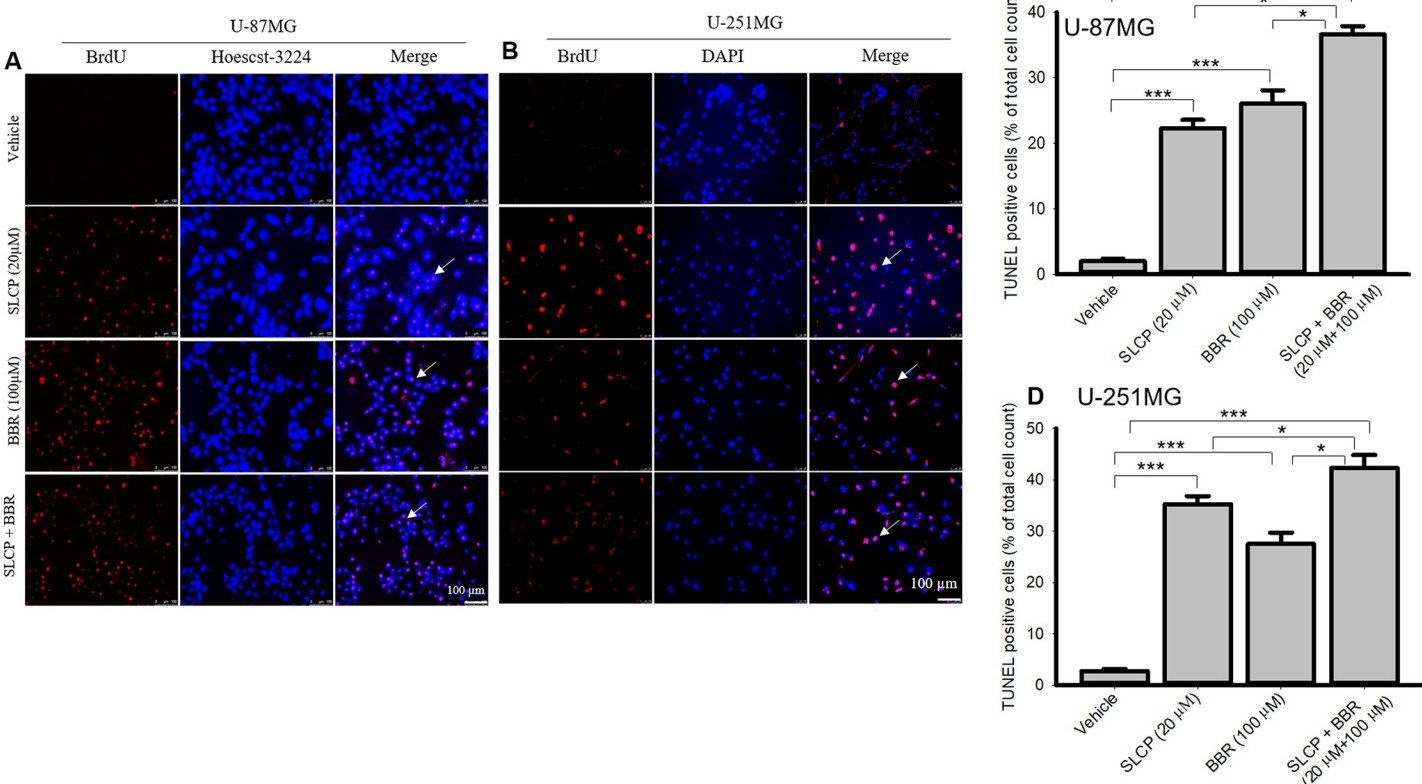

**Fig 3. More DNA fragmention of U-87MG and U-251 cells occurred after the combined SLCP and BBR therapy than by individual treatments.** U-87MG and U-251MG cells were grown in EMEM and they were treated with either SLCP (20 μM), BBR (100 μM) or their combination (in the 1:5 ratio) for 48 h. The cells were stained with terminal deoxyribonucleic acid nick end labeling (TUNEL) reagents for detecting DNA fragmented cells and counter-stained with Hoechst 33342. A fluorescent microscope (Leica Germany) was used to detect the signal with appropriate excitation/emission filters. A-B: Representative images of TUNEL/Hoechst 33342 stained U-87MG (A) and U-251MG (B) cells after treatment with SLCP, BBR or their combination. C-D: More TUNEL-positive cells were observed in combination-treated cells in both GB cell lines than for individual treatments. Arrows indicate TUNEL positive cells. Scale bars indicate 100 μm and is applicable to all images. *p<0.05 and ** p<0.01 and ***p<0.001 in comparison to vehicle-treated cells.

h of drug treatment (SLCP: 43.21%; BBR: 41.22%; SLCP+BBR: 60.72%; p<0.001) (**Fig 5A–5C**). Furthermore, to observe the nuclear morphology, we stained the cells with propidium iodide and Hoechst-33342. We observed a significant increase in the number of nuclear lobes in the SLCP+BBR-treated U-87MG cells than in SLCP- or BBR-treated cells after 48 h (p<0.05) (**Fig 5C and 5D**). In addition, we isolated the genomic DNA after the treatment of these drugs and observed smeared bands in drug-treated groups with ~1000-base-pair small DNA fragments in comparison to the vehicle-treated control U-87MG cells (**Fig 5E**). However, we did not observe any significant differences in DNA fragmented bands in either the co-treated and single drug-treated cells.

## 3.6. Combination treatment of SLCP and or BBR reduced mitochondrial markers and ATP levels in both GB cell lines more than did individual SLCP- or BBR-treatments

Decreases in mitochondrial membrane potential and ATP levels are critical signs of cell death. We measured mitochondrial health by staining both GB cell lines with JC-1, whereas ATP levels were measured by Glo-assays. We found significant decreases in the number of mitochondria (SLCP: 49.66%; BBR: 31.85%; SLCP+BBR: 75.87%) and the MMP in all the drug-treated cells, but a greater decrease was observed in the case of SLCP+BBR-treated groups (**Fig 6A**

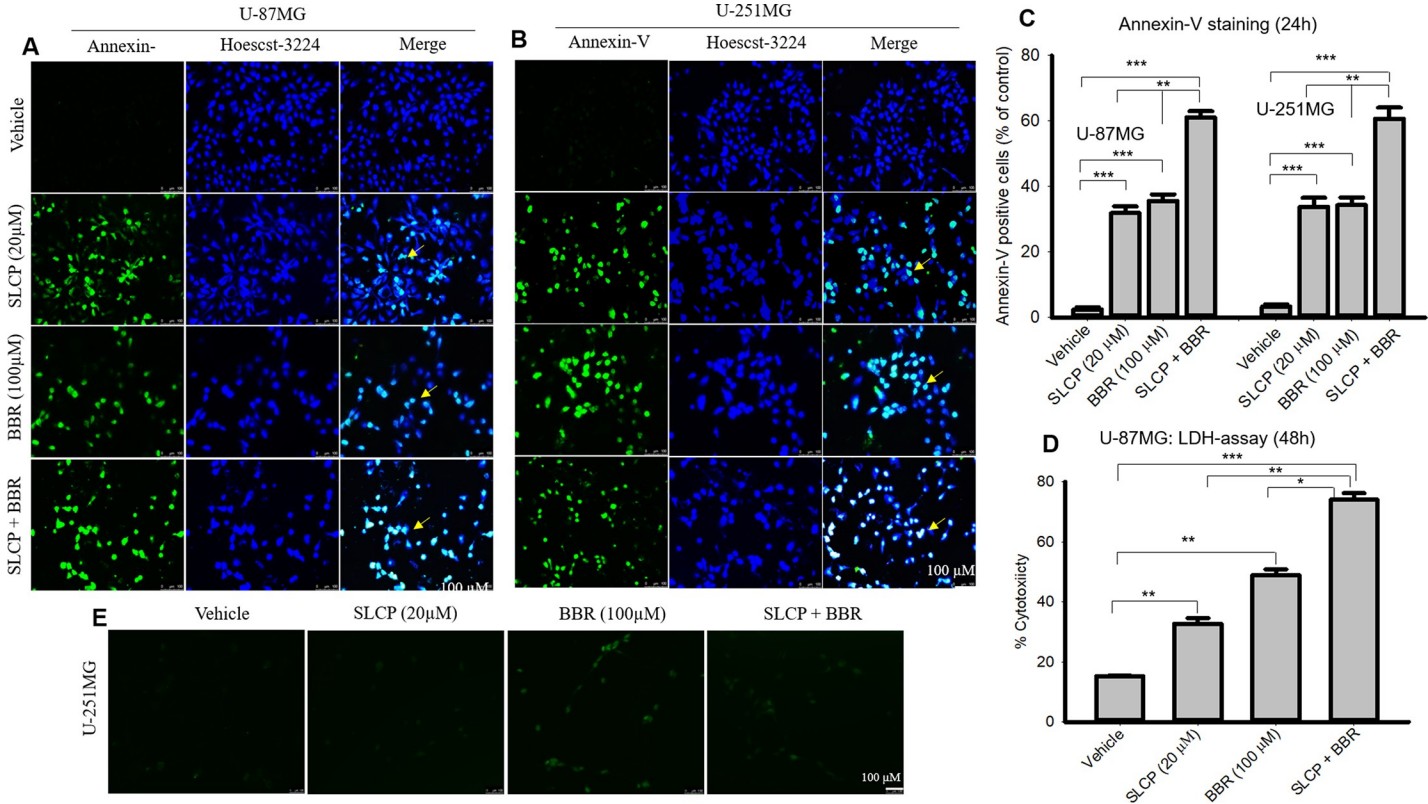

**Fig 4. Apoptotic death was greater in combination therapy with SLCP and BBR than their single treatments in both U-87MG and U-251 cells.** U-87MG and U-251MG cells were grown in EMEM and 1% pen/strep for 24 h, and then, the cells were treated with either SLCP (20 μM), BBR (100 μM) or combination (in a 1:5 ratio) for 24 h. The cells were stained with Annexin-V tagged with FITC for detecting apoptotic cell death and counter-stained with Hoechst-33342. Images were taken under a fluorescent microscope (Leica Germany) with appropriate excitation/emission filters. A-B: Representative images of Annexin-V/ Hoechst-33342-stained cells after treatment with SLCP or BBR or their combination for 24 h. C: More apoptotic cells were found in the combination-treated groups than in either the SLCP- or the BBR-treated cells. D: LDH release was increased in the combination-treated cells than those receiving individual treatments. Arrows indicate Annexin-V positive (apoptotic) cells. Scale bars indicate 100 μm and is applicable to all images. *p<0.05, ** p<0.01 and ***p<0.001.

and 6B). In addition, we performed Glo-assays, in which the luminescent signal is directly proportional to the level of available cellular ATP levels. We observed that the luminescent intensity (AU) was relatively less in the case of SLCP+BBR-treated cells in comparison to SLCP- or BBR-treated U-87MG cells (luminescent/ATP levels: SLCP = 11.77%/11.89%; BBR = 36.81%/37.18% and SLCP+BBR = 90.57%/91.48%) (Fig 6C and 6E). Furthermore, in the case of U-251MG cells, the luminescence and ATP levels showed similar trends to what was observed in the U-87MG cells. However, the percent change was less in the case of U-251MG cells compared to U-87MG cells (luminescent/ATP levels: SLCP = 11.84%/12.76%; BBR = 15.72%/16.95%; and SLCP+BBR = 8.00%/40.95%) (Fig 6D and 6F).

### 3.7. Reactive oxygen species (ROS) production was increased by combination of SLCP and BBR compared to individual treatments of SLCP or BBR

To investigate the cause of cell death after treatment with SLCP, BBR and their combination, we measured the ROS levels using the CellROX oxidative stress reagent. We observed that the ROS levels increased more in combination treatment of U-87MG cells as compared to individual treatment with SLCP or BBR (SLCP: 52.45%; BBR: 83.38%; and SLCP+BBR: 147.74%) (Fig

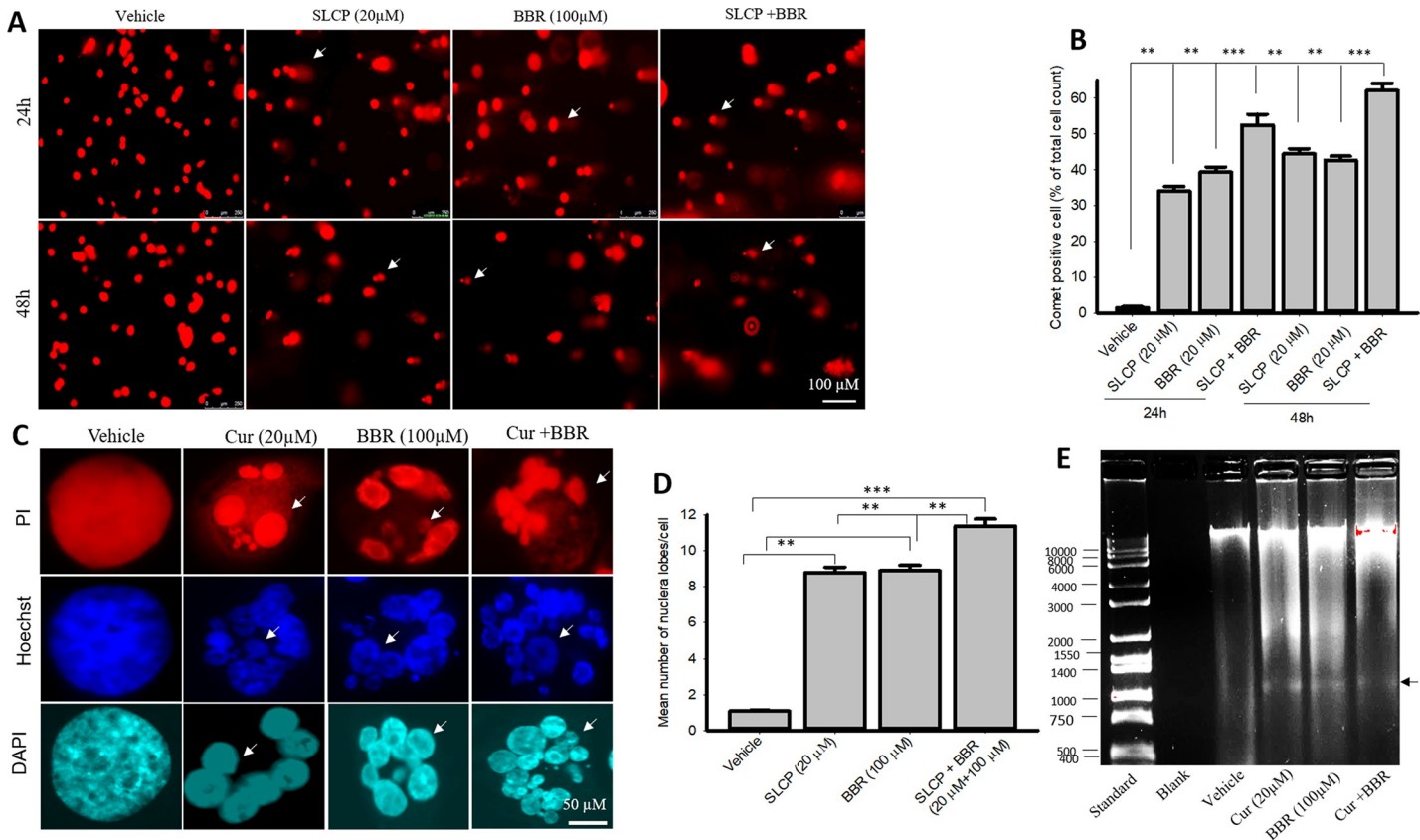

**Fig 5. Comet assay and nuclear morphology in U-87MG cells after treatment with SLCP, BBR, or their combination.** U-87MG cells were grown in EMEM and 1% pen/strep for 24 h and then were treated with either BBR (100 μM), SLCP (20 μM) or their combination for 24–48 h. Cells were lysed in lysis solution and, run in an electrophoretic chamber for 30 min, at which point the fragmented DNA tail was stained with ethidium bromide. The images were taken from a fluorescence microscope (Leica, Germany) with the appropriate filters. A-B: Representative images and their analyses showed that combination treatment significantly increased the fragmented DNA tail (comet) more than observed with single treatments. Arrows indicate comet-positive cells. Scale bar indicates 100 μm. $^{**}$p<0.01 and $^{***}$p<0.001. C-D: Nuclear morphology in U-87MG cells after treatment with SLCP or BBR or their combination. U-87MG cells were stained with propidium iodide and Hoechst 3342 and the images were taken with a fluorescence microscope (Leica Germany) using a 100x objective (total magnification 1000x). More nuclear lobes were observed in cells treated with both SLCP and BBR in comparison to those given single treatments. E: The gel electrophoresis image shows more DNA fragmentation in SLCP, BBR and their combination-treated group than in vehicle-treated cells. Arrows indicate fragmented nuclear lobe. Scale bars indicate 50- and 100 μm and are applicable to all images in their respective panels. $^{**}$p<0.01 and $^{***}$p<0.001.

7A and 7B). Similar phenomena were observed in U-251MG cells (SLCP: 28.87%; BBR: 39.72%; and SLCP+BBR: 48.51%) (**Fig 7A and 7C**). However, the degree of ROS production was less in U-251MG cells.

### 3.8. Combination treatment increased cell death markers and reduced cell survival markers more than individual treatments with either SLCP or BBR

We studied the cell death and cell survival markers from U-87MG and U-251MG cells after treatment with SLCP, BBR and their combination, to investigate the cell death mechanism. Our Western blot data (**Fig 8A–8E**) revealed that there was a stronger effect of the combined SLCP and BBR treatment on the levels of Bax (**Fig 8B**), Cyt-c (**Fig 8D**) and cleaved caspase-3 (**Fig 8E**) in both U-87MG and U-251MG cells. In contrast, Bcl$_2$ (**Fig 8C**) levels were significantly lower (p<0.05) in SLCP+BBR-treated cells in comparison to individual treatments of SLCP or BBR. Similarly, immunofluorescence intensity of cleaved caspase-3 (**Fig 8F and 8G**)

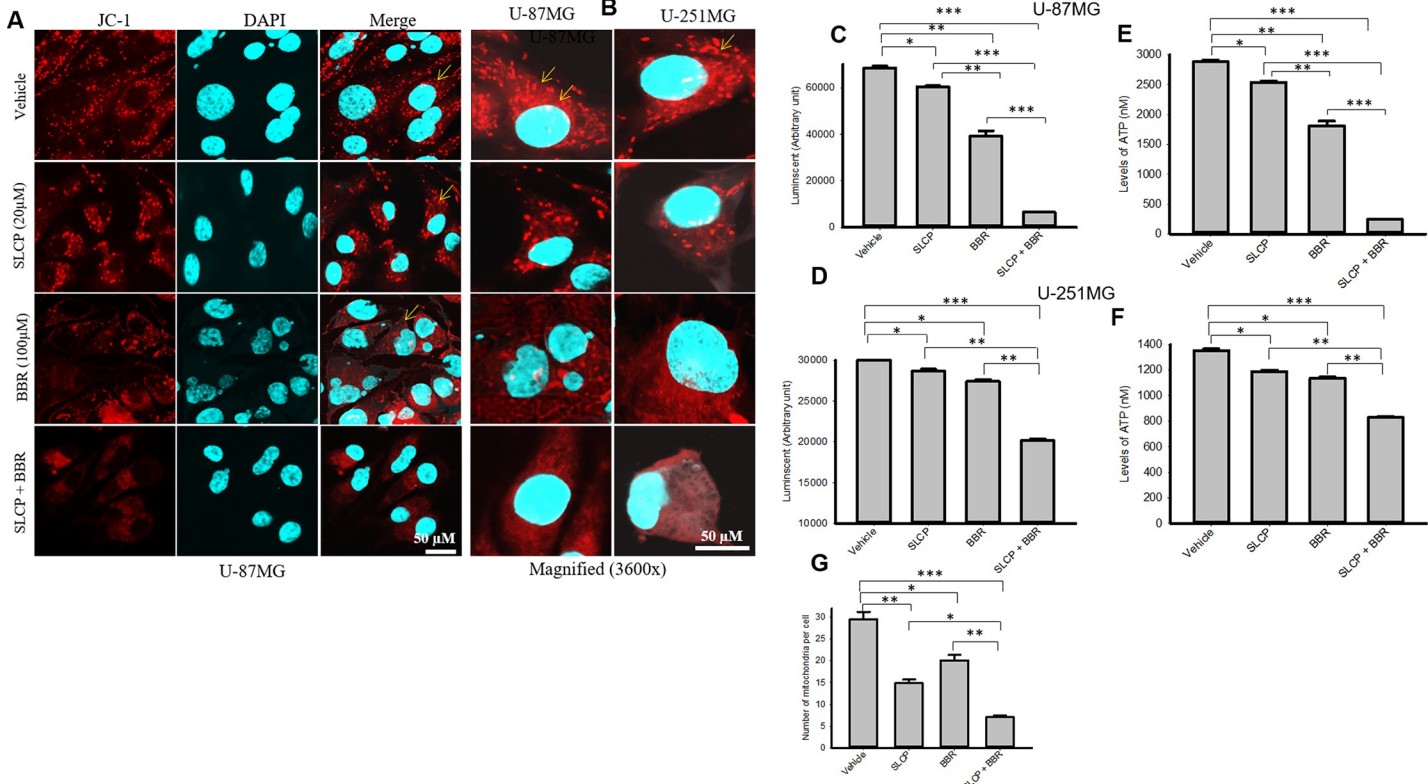

**Fig 6. Mitochondrial membrane potential and ATP levels were reduced more by combination therapy than by single treatments of SLCP or BBR in U-87MG and U-251MG cells.** U-87MG and U-251MG cells were grown in EMEM and 1% pen/strep for 24 h, and then, the cells were treated with either SLCP (20 μM) or BBR (100 μM) or their combination for 48 h at which point the cells were stained with JC-1 and images were taken using a confocal laser scanning microscope. **A**: Representative images showed a decrease in the number of mitochondria in SLCP- and BBR-treated cells, with combination treatments showing larger decreases. **B**: Higher magnification of U-87MG and U-251MG cells with JC-1 stain after treatment with SLCP, BBR or their combination. C-F: The Glo assay showed that there was a significantly lower luminescent signal and ATP levels in combination-treated U-87MG and U-251MG cells in comparison to their single-treated counterparts. Scale bars indicate 50 μm and are applicable to all images. *p<0.05 and **p<0.01 and ***p<0.001.

in U-87MG cells showed a greater increase in active caspase-3 in the SLCP+BBR-treated (p<0.01) cells in comparison to SLCP- or BBR-treated cells (SLCP: 86.26%; BBR: 73.44%; and SLCP+BBR: 205.63%).

## 3.9. p53 and c-Myc levels in U-87MG and U-251MG cells were modulated more by combination treatments than by single treatments

Western blot analysis from U-87MG cells revealed that c-Myc was significantly decreased by SLCP, BBR and SLCP+BBR treatment, but no significant differences were observed between treatment groups (**Fig 9A and 9B**). Although no major differences in c-Myc levels in U-251MG cells were observed, the p53 levels were significantly increased by SLCP+BBR-treated cells in U-87MG cells, in comparison to what was observed in SLCP or BBR-treated cells. In addition, our immunofluorescence data from U-87MG cells showed that c-Myc immunoreactivity was significantly decrease by SLCP+BBR-treated cells related to that in either SLCP- and or BBR-treated cells (SLCP: 23.95%; BBR: 40.52%; and SLCP+BBR: 58.25%) (**Fig 9D and 9E**). In addition, p53 immunoreactivity was significantly increased in cells given combination treatments, relative to their single-treatment counterparts (SLCP: 14.01%; BBR: 34.22%; and SLCP +BBR: 51.71%) (**Fig 9D and 9E**).

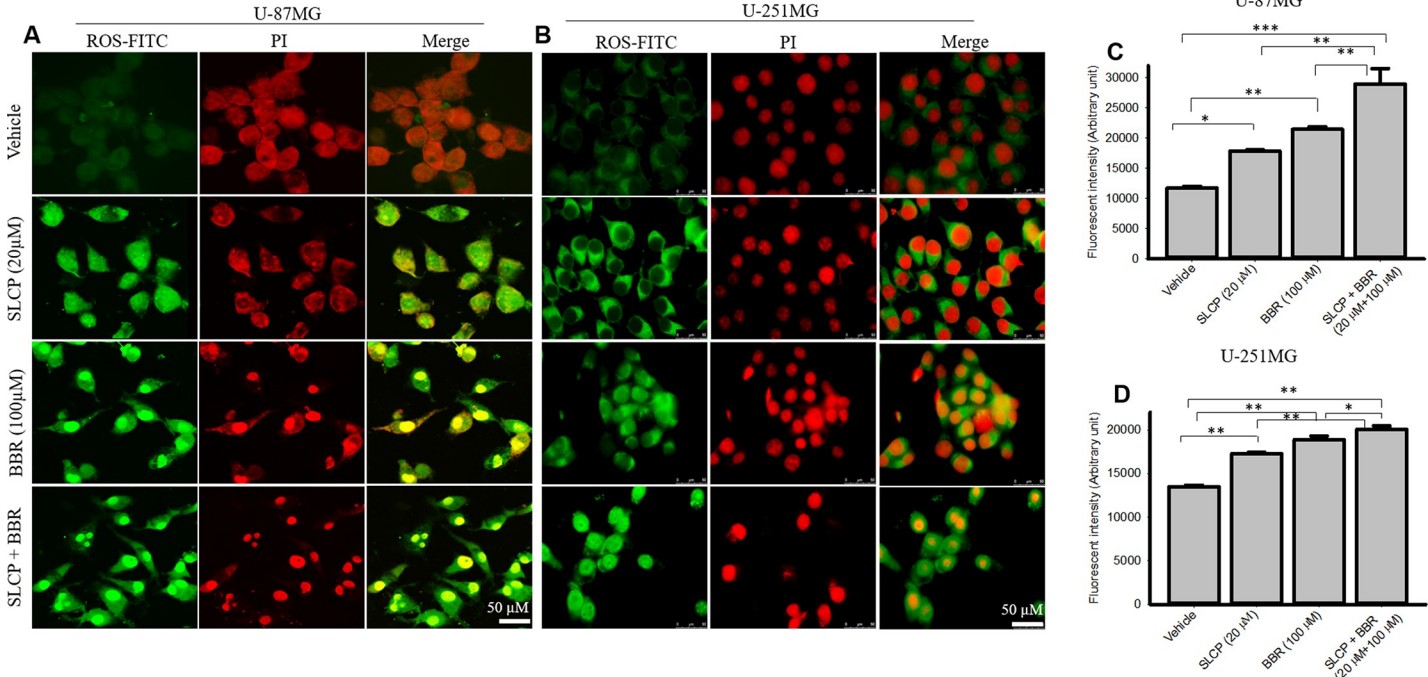

**Fig 7. Combination treatment of SLCP and BBR produced more ROS than did single treatments.** U-87MG and U-251MG cells were grown in EMEM and 1% pen/strep for 24 h and then treated with either SLCP (20 μM) or BBR (200 μM) or their combination for 48-h and leveled with CellROX reagent prior to counter-staining with propidium iodide. A-B: The images were taken using a confocal microscope (A) and a fluorescent microscope (B, Leica, Germany) with appropriate excitation/emission filters. The green fluorescent signal indicates ROS levels. C-D: Combination treatments increased ROS levels more than single treatments did. Scale bar indicates 50 μm and is applicable for all the images. $^{*}p<0.05$, $^{**}p<0.01$ and $^{***}p<0.001$.

### 3.10. Combination treatment of SLCP and BBR inhibited the PI3K/Akt/mTOR pathway in U-87MG and U-251MG cell lines more efficiently than did single treatments

The PI3K/Akt/mTOR pathways play a pivotal role in GB growth and proliferation. Our Western blots analyses showed that p-Akt (In U-87MG cells: SLCP = 19.31%; BBR = 26.19%; and SLCP+BBR = 70.71% and for U-251MG cells: SLCP = 19.20%; BBR = 52.34%; and SLCP+BBR = 89.91%) and total Akt (In U-87MG cells: SLCP = 19.44%; BBR = 31.40%; and SLCP+BBR = 45.55% and for U-251 cells: SLCP = 27.76%; BBR = 18.66%; and SLCP+BBR = 33.63%) were significantly downregulated more for SLCP+BBR-treated cells than for SLCP- or BBR-treated cells (**Fig 9A–9C**).

Similarly, p-PI3Kp85 was significantly reduced in both the GB cell lines after treatment with SLCP or BBR. However, a greater decrease was noted in the case of co-treatment (in U-87MG cells: SLCP = 33.38%; BBR = 26.11%; and SLCP+BBR = 84.50% and for U-251MG cells: SLCP = 42.84%; BBR = 47.63%; SLCP+BBR = 42.04%). Total PI3Kp85 (in U-87MG cells: SLCP = 33.38%; BBR = 26.11%; and SLCP+BBR = 53.85%) was also reduced more in co-treated cells than in their single-treated in U-87MG cells. However, we did not observe such changes in the case of U-251MG cells (SLCP = 14.41%; BBR = 9.42%; and SLCP+BBR = 4.56%) (**Fig 9A, 9D and 9E**).

In addition, we have also investigated the total mTOR and p-mTOR levels after treatments with SLCP or BBR and their combination. Our Western blot analysis showed that total mTOR levels were significantly decreased in SLCP+BBR-treated cells more than in their single treatment counterparts in both GB cell lines [in U-87MG cells: SLCP = 18.86%; BBR = 25.55%; and

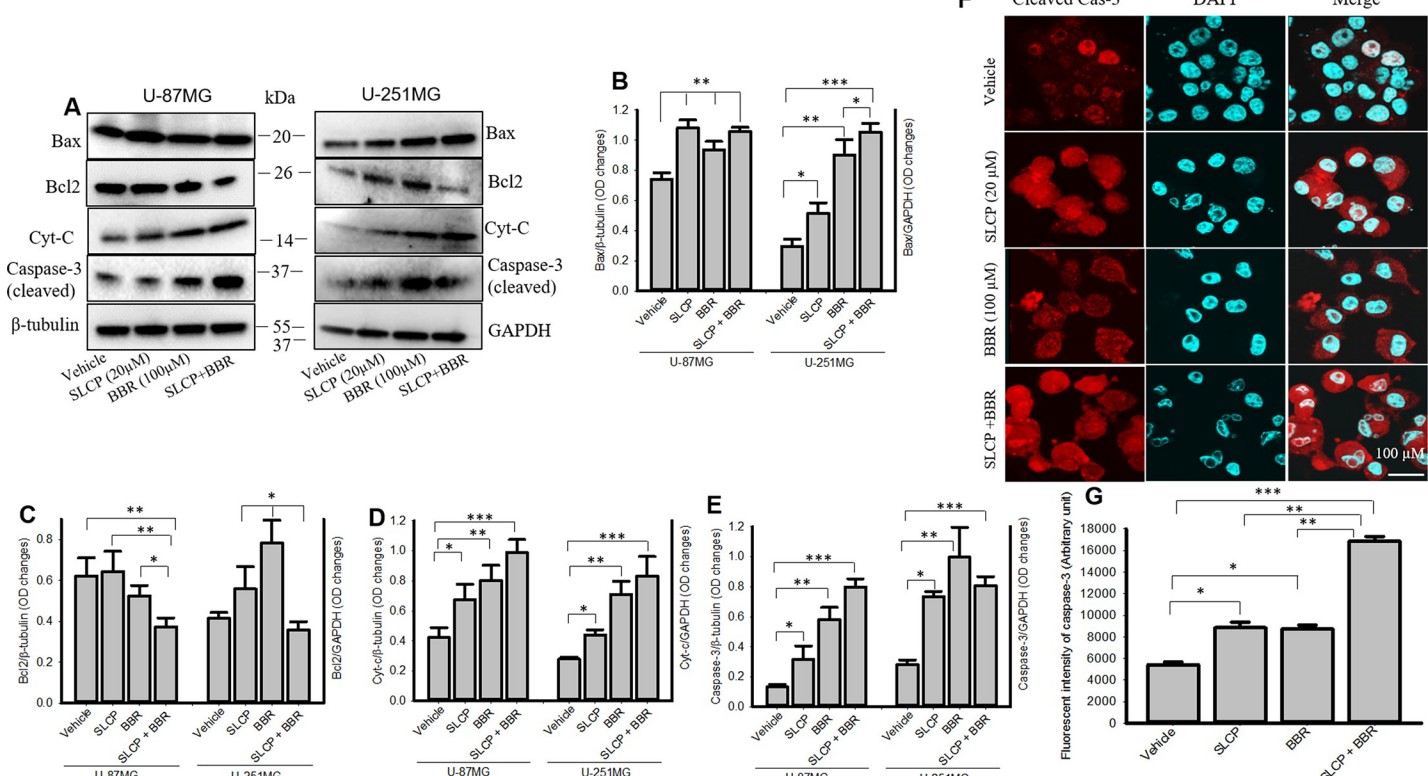

**Fig 8. Combination treatment induced more cell death and decreased cell survival markers more than did single treatments.** A-F: Cell death markers, such as Bax, cytochrome-c and cleaved caspase-3 were significantly increased in SLCP + BBR-treatments in both U-87MG and U-251MG cells, relative to single-treatments of SLCP or BBR alone. Cell survival markers, such as Bcl2, were significantly reduced in SLCP+BBR-treated cells more than those by individual SLCP and BBR treatments. F-G: Immunocytochemistry of cleaved caspase-3 in U-87MG was significantly increased in SLCP+BBR-treated cells, relative to SLCP- or BBR-treated cells. Scale bar indicates 100 μm and is applicable to other images. *p<0.05, **p<0.01 and ***p<0.001.

SLCP+BBR = 42.61% and for U-251 cells, in which SLCP (34.99%) and BBR (42.37%) showed increases, but SLCP+BBR did not (61.53%)] (**Fig 10A and 10F**). Similarly, we also observed more significant decreases in levels of p-mTOR after co-treatment than with either SLCP- or BBR-treatments in U-87MG cells (in U-87MG cells: and SLCP = 27.30%; BBR = 62.21%; SLCP + BBR = 73.42%). Although the mTOR levels were lower in SLCP- or BBR-treated or co-treated U-251MG cells in comparison to vehicle-treated cells, we did not observe differences between co-treatment and SLCP and or co-treatment and BBR-treated cells (SLCP = 21.12%; BBR = 54.77%; SLCP+BBR = 52.59%) (**Fig 10A and 10G**).

## 4. Discussion

Targeting glioblastoma using natural polyphenols is a promising strategy due to their anti-cancer and anti-inflammatory properties [46–48]. Curcumin (Cur) and berberine (BBR) have been shown to be potent anti-carcinogenic natural compounds for several types of cancer [17, 29, 49]. In the present study, we have compared the synergistic, anti-cancer effects of permeable, solid-lipid Cur particles (SLCP) and BBR on two different GB cell lines that were derived from human tissue (U-87MG and U-251MG). We found that co-treatment of SLCP and BBR induced more DNA fragmented cells, increased reactive oxygen species (ROS), induced mitochondrial dysfunction, reduced the number of mitochondria per cell, significantly declined of ATP levels, induced more cell death markers, increased apoptotic death and inhibited the

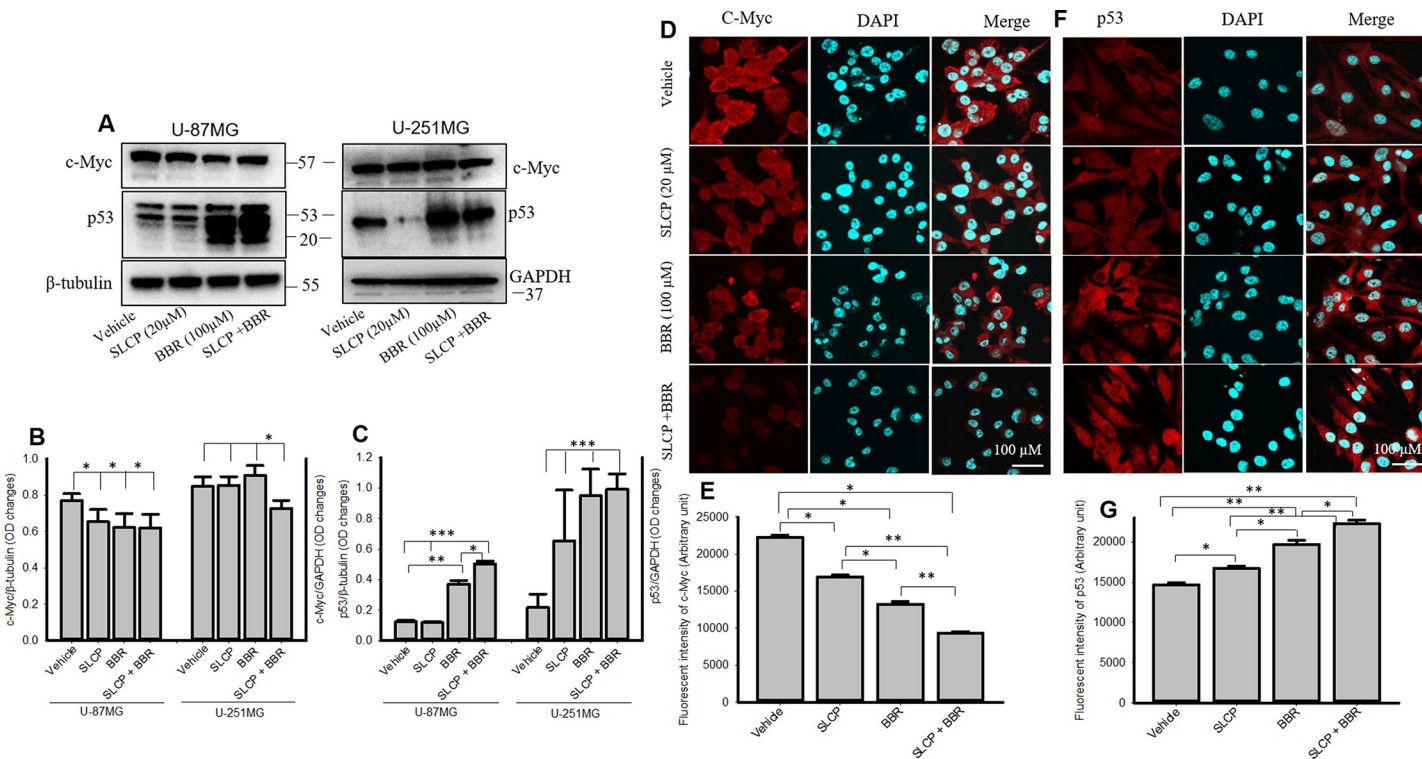

**Fig 9. Combination treatments increased p53 and decreased c-Myc levels in U-87MG and U-251MG cells more than did single treatments.** A-C: Western blot data showed that SLCP+BBR increased p53 and decreased c-Myc levels significantly more in U-87MG and U-251MG cells, in comparison to single treatments. Similarly, immunofluorescence signals of c-Myc was decreased (D-E) and p53 was increased (F-G) more in SLCP+BBR-treated U-87MG cells than in SLCP- or BBR-treated cells. Scale bar indicates 100 μm and is applicable to other images. *p<0.05, **p<0.01 and ***p<0.001.

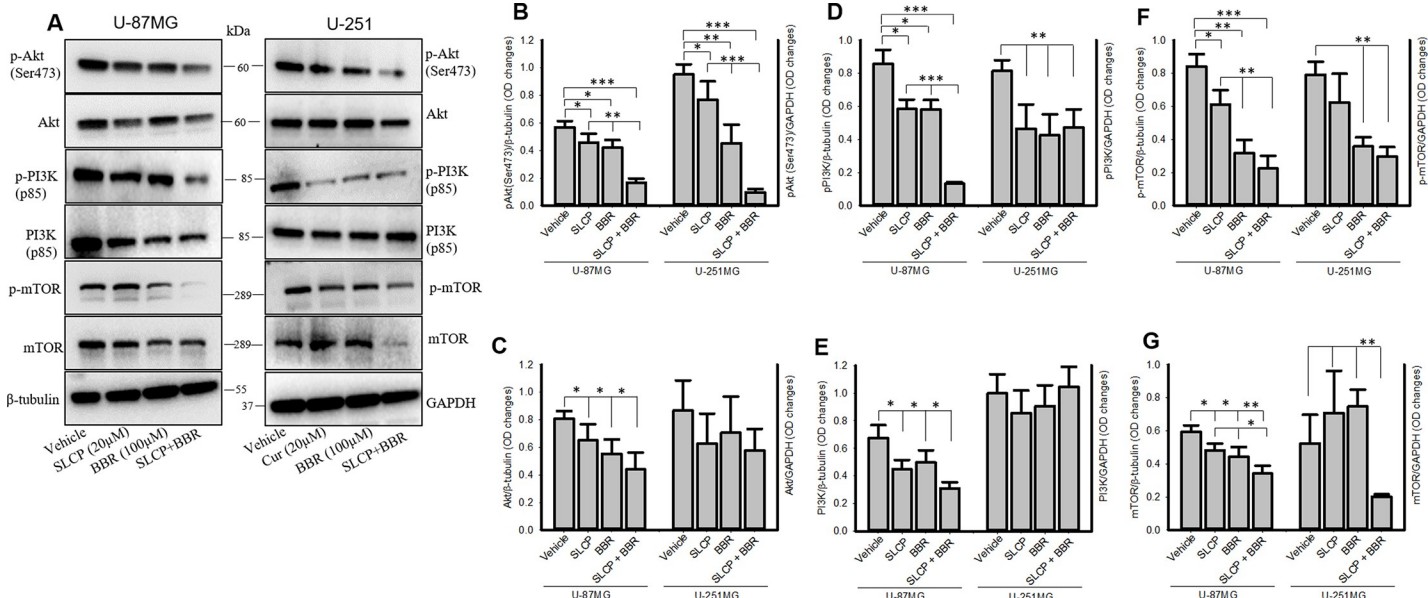

**Fig 10. Combination treatment decreased PI3K/Akt/mTOR pathways more efficiently than SLCP or BBR treatments alone.** U-87MG and U-251MG cells were grown in EMEM and 1% pen/strep for 24 h and then the cells were treated with either SLCP (20 μM) or BBR (100 μM) or their combination (in this 1:5 ratio) for 24 h and then cells were extracted, and Western blots were performed and probed with different antibodies. A-F: Western blot data showed that SLCP+BBR inhibited the PI3K/Akt/mTOR pathways more efficiently than either SLCP- or BBR-treated cells. *p<0.05, **p<0.01 and ***p<0.001.

PI3K/Akt/mTOR pathway more efficiently than either single treatment. Overall, our data demonstrated that the co-treatment of SLCP and BBR are more potent to kill cultured GB cells than either single treatment.

The major limitation of FDA-approved chemotherapeutics, such as temozolomide (TMZ), in treating GB is that it induces neuroinflammation, and develops clinically significant toxicity [50]. In addition, GB becomes resistant to TMZ following its repeated use [51, 52]. To overcome these issues, the combined treatment of anti-cancer, as well as anti-inflammatory drugs could be a better candidate to prevent GB growth and proliferation. Recent experimental evidence suggest that natural anti-cancer agents could be used as a promising strategy to reduce inflammation, prevent tumor invasion and metastasis [46–48]. In this context, Cur and BBR have exhibited potent anti-cancer activities in several malignancies, including breast cancer [29], colon, and brain tumors [1, 25]. However, due to the poor solubility and less chemical stability in different body fluids, the anti-cancer capacity of either Cur [53] or BBR [54] as individual treatments is limited. We have been using a lipid formula of Cur (SLCP) to increase Cur stability and bioavailability [35, 55–57]. Recently, we have shown the greater anti-cancer activities and induction of autophagy by SLCP in cultured GB cells than what is observed with natural Cur [28, 58], reducing the amount of Cur needed to be effective [53]. Given the efficacy of both SLCP and BBR, especially when given together, in reducing other types of cancers, we hypothesized that the combination of SLCP and BBR would have similar additive effects against GB [29]. In this context, Liu and colleagues reported that BBR induces senescence of human GB cells by downregulating the EGFR–MEK–ERK signaling pathway [59]. Similarly, the synergistic chemopreventive effects of Cur and BBR have been reported by Wang and colleagues on human breast cancer cells [29]. In addition, Balakrishna and colleagues also observed a synergetic anti-cancer activity of BBR and Cur on different cancer cells, such as liver, breast, lung, bone, glioma, and leukemia cell lines [25, 30]. Another study conducted by Yin and colleague reported that Cur sensitizes GB to TMZ by simultaneously generating ROS and disrupting AKT/mTOR signaling [7]. These findings prompted us to investigate the potential co-treatment effects of Cur and BBR against GB.

In the present study, we have compared the mechanism of cell death by investigating the PI3K/Akt/mTOR pathway after single or co-treatment of SLCP and BBR in two different GB cell lines (U-87MG and U-251MG) derived from human tissue. We have performed several cell viability and cell death assays to confirm their synergistic anti-cancer effects. After performing dose-response studies with SLCP and BBR, we decided to use a one-parts of SLCP and five-parts of BBR for further experiments; because 20 μM of SLCP and 100 μM of BBR showed optimum cell death (**Fig 1**). Further, to assess whether SLCP or BBR selectively induced GB cell death without affecting neuronal death, human neuroblastoma (SH-SY5Y) and mouse neuroblastoma (N2a) cells were treated with the same concentrations of SLCP (20 μM), and BBR (100 μM) and their combination (1:5 ratio). We observed only 5–10% cell death in both SLCP and or BBR treated cells, indicating Cur and BBR had a minimal effect on neuroblastoma. The co-treatment in N2a caused a 10–20% cell death (**Fig 1D**), whereas the co-treatment group in SH-SY5Y produced around 30–40% cells death, indicating combination treatment can have toxic effects on human neuroblastoma (**Fig 1E**), but in a different manner than observed in GB cell death, suggesting that the mechanisms are probably different as well [8, 60–62]. The cell viability (MTT assay) and cell death (LDH assay) data confirmed that co-treatment of SLCP and BBR has greater toxic effects on U-87MG and U-251MG cells (**Fig 1**) than either single treatment, as reported previously [29, 30]. In addition, we have also performed cell proliferation assays after separate and co-treatments of these drugs, and our results clearly show greater inhibition of cell proliferation in the case of co-treated cells (**Fig 2**),

indicating an additive effect of these drugs on GB proliferation, as reported by other investigators, using, in different cancer cells [29, 30].

To understand the type of cell death, we have performed TUNEL and Annexin-V staining, which identifies the DNA-fragmented cells and apoptotic death, respectively [63]. We found many more TUNEL-positive cells in the case of the co-treatment group in both U-87MG and U-251MG cells (**Fig 3**), which correlated with our MTT, LDH, and cell proliferation assays data (**Figs 1 and 2**). Further, to confirm the mode of cell death, we performed Annexin-V staining, and found more Annexin-V positive cells in the co-treatment group (**Fig 4**) for both lines of GB cells, which correlates with the TUNEL staining data (**Fig 3**). We have used Annexin-V-FITC staining kit to detect apoptotic death which produce green fluorescent signal, whereas both BBR and SLCP treatment also might produce green fluorescent signal which could interefere with Annexin-V staining. Therefore, we imaged the cells exactly same way after treatment with BBR and SLCP. We observed a weak green signal produced by BBR and or SLCP (**Fig 4E**), however the green fluorescent signal produced by BBR and or SLCP was negligible in comparison to Annexin-V positive cells. Although Annexin-V with red fluorescen dye might be used to visualize apoptotic cells more efficiently than does Annexin-V-FITC after SLCP or BBR treatment.

The current study also investigated the morphology of nuclei using three different dyes: propidium iodide (PI), Hoechst-33342 and DAPI; the number of fragmented nuclear lobes were counted. We found that the co-treatment group showed more nuclear lobe formation in comparison to SLCP- or BBR-treated cells after 24 h (**Fig 5**), which, again, confirms that co-treatment induces greater nuclear damage than individual treatments of those drugs. Furthermore, we also performed comet assays, a gold standard for the studying the degree of DNA fragmentation *in vitro* [64]. More comet positive cells were observed in both GB cell lines in the case of the co-treatment group (**Fig 6**), which correlated with the TUNEL staining data. These data, again, confirmed the greater induction of cell death by combination therapy than by individual SLCP or BBR treatments. Although we did not analyze the head and tail fluorescent intensity, fragmented DNA tail length and tail-moment length gave a full profile of the degree of DNA fragmentation after drug treatment. However, the number of comet-positive cells correlated with TUNEL-positive and with Annexin-V cells in both the GB cell lines. This indicates that the combination treatment is more efficient in damaging the DNA than what is observed with single treatments. In addition, we performed DNA gel electrophoresis after 24 h of SLCP and or BBR and their co-treatment in U-87MG cells. We observed smeary, fragmented DNA bands (**Fig 5E**), which confirmed that both the drugs induced DNA fragmentation in GB cells.

Many events and signaling pathways are involved in DNA fragmentation and apoptosis following treatments with SLCP, BBR or their co-treatment (**S1 Fig**). One of the key events is mitochondrial damage. Due to its crucial role in cell death, mitochondrial damage has recently been characterized as the "motor of cell death". We tried to correlate apoptosis with mitochondrial bioenergetic signaling following treatment with SLCP or BBR. For example, we have investigated mitochondrial health by performing JC-1 staining, an indicator of mitochondrial membrane potential (MMP) [65]. JC-1 is more specific for MMP and more consistent in its response to depolarization than classical mitochondrial dyes, such as DiOC6(3) and rhodamine 123 [66]. The ratio of green to red fluorescence of JC-1 depends on MMP and not on mitochondrial size, shape, and density [65]. Therefore, decrease in the red fluorescence signal indicated mitochondrial degeneration or cell death which we observed after treatment with SLCP and BBR (**Fig 6A and 6B**). Further, decreases in mitochondrial number (**Fig 6G**) in the treatment groups indicate that these drugs exerted deleterious effects on mitochondrial survivability, suggesting overall poor health of the mitochondria. Similarly, we also performed Glo-

assays, which measure the cellular ATP levels. We clearly observed a larger decrease in the number of mitochondria and mitochondrial function in the co-treatment groups (**Fig 6C–6F**), in comparison to single treatments of SLCP or BBR, suggesting an additive deleterious effect on mitochondria. We also investigated the probable causes of mitochondrial degeneration and decreased levels of ATP after Cur and or BBR treatment. Increased levels of ROS is a major causative factor in cell death after treatment with anti-cancer drugs. Therefore, we measured the total ROS levels after treatment with SLCP, BBR, and their combination [7, 9], and we observed that co-treatment induced greater levels of ROS (**Fig 7**). However, several biochemical techniques are available to detect different species of ROS. Increased levels of ROS involve releasing cell death markers from the mitochondria, such as what occurs when using apoptotic-inducing factor (AIF), Bax and Cyto-c; it also activates caspase-3 [67]. For example, we observed a significant upregulation of Bax, Cyt-c and active caspase-3 levels in co-treated GB cell lines, compared to what was observed following single treatments, suggesting that the combination treatment has stronger effects on cell death and mitochondrial degeneration. In contrast, downregulation of the Bcl$_2$ protein by co-treatment was observed, relative to individual treatments with SLCP or BBR, suggesting combination treatment can induce more apoptotic cell death than single treatments, which is supported by the work of other investigators [29, 30]. Therefore, decreases MMP and ATP levels caused by SLCP or BBR treatment may be due to an increased generation of ROS, which could initiate DNA fragmentation and apoptotic death (**S1 Fig**).

Tumor suppressor genes, such as p53, as well as oncogenes, such as c-Myc, play important roles in regulating GB development. Downregulation of p53 increases the susceptibility for tumor formation, whereas its upregulation can prevent tumor formation [68]. When we treated GB cells with SLCP or BBR, we observed an increase in levels of p53, indicating anti-tumor activity, whereas co-treatment further upregulated p53 levels, indicating an even stronger effect on this protein (**Fig 9A and 9B**). However, we observed levels of p53 in U-87MG cells were different from U-251MG cells. The discrepancy might be due to the way the total amount of protein was loaded. For example, in the case of U-251MG cells, the amount of loaded protein was one-third of the protein loaded by U-87MG cells. C-Myc is another oncogenic marker in cell [69], and its activation leads to the upregulation of many genes which are involved in cell proliferation and cancer development [69]. We found a significant decline of c-Myc levels in SLCP- and BBR-treated U-87MG cells (**Fig 9C and 9D**) and their co-treatment decreased this level even more in the case of U-251MG cells (**Fig 9C and 9D**), which, again, confirms that co-treatment of SLCP and BBR has greater anti-proliferative and anti-carcinogenic effects than individual treatments. Previously, we reported an increase in p53 and decrease in c-Myc in cultured GB cells after treatment with natural Cur and SLCP (25 μM), which was confirmed by the present findings [55].

The mammalian target of rapamycin (mTOR), is a serine/threonine kinase of the PI3K (phosphoinositide 3-kinases)-related kinase family involve in cancer cell growth and cell survival [70]. These proteins have a strong links in the formation of the tumor microenvironment and drug resistance. Therefore, blocking this pathway can reduce proliferation, migration and survival of cancer cells [23]. Several natural compounds, such as BBR, resveratrol, Cur, quercetin and others, can modulate the mTOR pathway [49]. Vengoji and colleagues recently reported that BBR has anti-tumor effects through inhibition of the mTOR-signaling pathway and can induce senescence of human glioblastoma cells by downregulating the EGFR–MEK–ERK signaling pathway [41]. In another study, Wang and colleagues reported that BBR induced autophagy by inhibiting the AMPK/mTOR/ULK1 (Unc-51-like-autophagy-activating kinase) pathway [25]. Similarly, glucose uptake is also reduced by BBR, via inhibition glucose transporter 1 (GluT1), lactate dehydrogenase-A (LDH-A), and hexokinase-2 (HK-2)

expression, which induces apoptosis and thus inhibits tumor growth and invasiveness [25]. BBR can also induce autophagy by activation of Beclin-1 and inhibition of mTOR signaling by suppressing the activity of Akt and up-regulating that of P38-MAPK signaling [35,36]. Interestingly, synergistic anti-tumor activities of rapamycin and BBR have been observed in hepato-carcinoma cell lines [38]. Similarly, we and others have reported that Cur is a potent inhibitor for the PI3K/Akt/mTOR pathway in diverse types of cancers, including GB [28]. Therefore, we sought to further inhibit the PI3K/Akt/mTOR pathway by co-treatment of BBR and SLCP. Our Western blot data revealed that co-treatment of SLCP and BBR inhibited PI3K, Akt and mTOR and their phosphorylation levels more than their individual treatments (Fig 10), suggesting that the co-treatment is more effective in blocking the PI3K/Akt/mTOR pathway and GB progression than using single treatments of these drugs. Although we found the synergistic anti-cancer effects of SLCP and BBR in cultured GB cell lines, the mechanistic details of GB growth, invasion, proliferation, malignancies, and metastasis in animal or human brain are much more complex than in GB cell culture models. Therefore, a more complete understanding of the mechanisms of both the individual and combined effects of SLCP- and BBR-induced GB cell death will require further experiments in animals to further confirm and optimize this therapeutic strategy prior to its clinical use.

## Conclusions

Overall, we report that co-treatment of anti-cancer natural polyphenols, such as Cur and BBR, have higher inhibitory effects on GB growth and proliferation than do individual treatment. Both SLCP and BBR synergistically damage mitochondria and decreased ATP levels, along with inducing DNA fragmentation and apoptotic death in both GB cell lines than single treatment. The combination of SLCP and BBR increased ROS levels and induced cell-death markers greater than produced by single treatments with these drugs. Furthermore, co-treatment of SLCP and BBR effectively inhibited the PI3K/Akt/mTOR signaling pathway more than did treatments of SLCP or BBR alone. Taken together, our findings suggest that co-treatment of SLCP and BBR may be a promising therapy for GB.

## Supporting information

**S1 Fig. Schematic diagram showing the mechanism of action of curcumin and or Berberine in GB cell death.**
(TIF)

**S2 Fig. Raw data for all gels and Western blots.**
(PDF)

**S3 Fig. Raw data for all data analyses.**
(XLSX)

**S4 Fig. All Western blot data for U-87MG cells.**
(XLSX)

**S5 Fig. All Western blot data for U-251MG cells.**
(XLSX)

## Author Contributions

**Conceptualization:** Panchanan Maiti.

**Data curation:** Panchanan Maiti, Alexandra Plemmons.

**Formal analysis:** Panchanan Maiti.

**Funding acquisition:** Gary L. Dunbar.

**Investigation:** Panchanan Maiti.

**Methodology:** Panchanan Maiti.

**Project administration:** Panchanan Maiti.

**Resources:** Panchanan Maiti.

**Software:** Panchanan Maiti.

**Supervision:** Panchanan Maiti.

**Validation:** Panchanan Maiti.

**Visualization:** Panchanan Maiti, Gary L. Dunbar.

**Writing – original draft:** Panchanan Maiti.

**Writing – review & editing:** Panchanan Maiti, Alexandra Plemmons, Gary L. Dunbar.

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
