## [Decision Letter · Decision Letter 0]

27 Sep 2019

PONE-D-19-22431

Combination treatment of Berberine and solid lipid curcumin particles increased cell death and inhibited PI3K/Akt/mTOR pathway of human cultured glioblastoma cells more effectively than did individual treatments

PLOS ONE

Dear Dr. Maiti,

Thank you for submitting your manuscript to PLOS ONE. After careful consideration, we feel that it has merit but does not fully meet PLOS ONE’s publication criteria as it currently stands. Therefore, we invite you to submit a revised version of the manuscript that addresses the points raised during the review process.

Please address the reviewer's points.

We would appreciate receiving your revised manuscript by Nov 11 2019 11:59PM. To enhance the reproducibility of your results, we recommend that if applicable you deposit your laboratory protocols in protocols.io, where a protocol can be assigned its own identifier (DOI) such that it can be cited independently in the future. For instructions see: http://journals.plos.org/plosone/s/submission-guidelines#loc-laboratory-protocols

We look forward to receiving your revised manuscript.

Kind regards,

Ferenc Gallyas, Jr., Ph.D., D.Sc.

Academic Editor

PLOS ONE

Journal Requirements:

3. We noticed minor instances of text overlap with the following previous publication(s), which need to be addressed:

https://www.hindawi.com/journals/omcl/2017/9656719/

The text that needs to be addressed involves the Methods and Results section.

In your revision please ensure you cite all your sources (including your own works), and quote or rephrase any duplicated text outside the methods section. Further consideration is dependent on these concerns being addressed.

Additional Editor Comments (if provided):

Reviewers' comments:

Reviewer's Responses to Questions

**Comments to the Author**

1. Is the manuscript technically sound, and do the data support the conclusions?

Reviewer #1: Partly

2. Has the statistical analysis been performed appropriately and rigorously? 

Reviewer #1: Yes

3. Have the authors made all data underlying the findings in their manuscript fully available?

Reviewer #1: Yes

4. Is the manuscript presented in an intelligible fashion and written in standard English?

Reviewer #1: Yes

5. Review Comments to the Author

Reviewer #1: On the whole, the work reported in the manuscript includes significant new results and represents an interesting contribution to the understanding of anti-tumor properties of natural compounds. Hence, I think that the results presented here have enough interest to be published in this Journal.

However, I have a number of reservations and comments that need to be addressed before this manuscript can be accepted:

I the issues I found in this context, as it follows below (the sentences/words of the manuscript are reported in italics”:

-Firstly, the Authors should carefully check the text for inaccuracies and typing mistakes.

-The Authors should report the statistical test they used to verify normality and omoschedasticity of data before applying ANOVA.

-page 1

Title: berberine not Berberine

-page 2:

Abstract: In my opinion, the “synergistic effect” term has been used improperly and therefore should be replaced with “higher effect” or “lower effect”. Similarly, in all text synergy-derivative terms have to be corrected. In fact, a synergistic effect should be properly demonstrated and mathematically calculated (for example by using EC50).

Abstract: “ascess” to be replaced with “assess”.

-page 3

Introduction: “It exhibits anticancer activity in glioma, colorectal-, lung-, prostate- and ovarian cancer, by inducing apoptotic cell death (20)”

I think that other specific references should be added to (20).

Introduction: “Recently, Wang and colleagues reported that BBR induces autophagy in GB by

targeting the AMPK/mTOR/ULK1-pathway (21).”

A recent paper (Agnarelli et al., 2018 Scientific Reports) showed that berberine induced autophagy in U343 GB cells. For this reason, I would suggest to cite this reference in the text, as an example:

"It has been demonstrated that BBR induces autophagy in different GB cell lines U87, U251, U343 (21; Agnarelli et al., 2018). In particular, Wang and colleagues reported that BBR induces autophagy in U87 and U251 cells by targeting the AMPK/mTOR/ULK1-pathway (21).

Introduction: “Similarly, Jin and colleagues reported that BBR inhibits angiogenesis in GB xenografts”.

I suggest to replace “Similarly” with “In addition”.

-page 4

Introduction: “Recently, we have shown that by using solid lipid curcumin particles (SLCPs), stronger anti-cancer effects and inhibition of the PI3k/Akt/mTOR pathway in human GB cell were observed than when natural Cur was used [20]”

In the text the Authors wrote:"we have shown" but the reference (20) is Meeran et al., (2008).

-page 6

Materials and methods. 2.3: “whereas combination treatment of SLCP and BBR was 1:5 (20 μM + 100 μM).”

The Authors should explain/comment why they choosed the combination with 1:5 ratio SLCP:BBR

and analyzed the effects of this combination.

-page 7

Materials and methods. 2.7: “The total number of cells and the number of annexin-V-positive cells (green) were counted per microscopic field and expressed as a percentage of dead cells.”

As BBR emits fluorescent green light when appropriately excited, I suggest to use Annexin V red fluorescent staining in order to analyze apoptotic cell death.

-page 7

Materials and methods. 2.9: In order to assess the impact of treatments on intracellular ROS level, I suggest to use a different method based on measurement of the fluorescence intensity using a Multilabel Plate Reader.

-page 9

Results 3.3: “in U-87MG cells (Fig 2A and C)” have to be replaced with “in U-87MG cells (Fig. 3A and C)”

-page 9

Results 3.3: “of their treatment (Fig 2A and C)” have to be replaced with “of their treatment (Fig. 3B and D)”

-page 10-11

Results 3.6: “We measured MMP by staining both GB cell liness with JC-1”

In my opinion, JC-1 should be used as a mitochondrial marker not as an MMP indicator (in fact the Authors did not show MMP measurements in Results). I suggest to use a biochemical method to measure a possible mitochondrial dysfunction.

As a minor correction, “liness” have to be replaced with “lines”.

-page 11

Results 3.6: (Graph D and F) have to be corrected with (Fig. 6D and F).

-page 12

Results 3.9: The p53 pattern is composed of different size bands. In addition the p53 expression pattern observed in the two glioblastoma cell lines is different. The Authors should comment these observations.

-page 13-19

Discussion, Fig. 11 and Conclusion have to be modified accordingly to suggestions.

-Figure 1A

Piknotic nuclei are not clearly visible.

The authors should show higher magnified images in order to demostrate the presence of piknotic cells.

-Figure 4D

LDH-related results are not described in the text

-Figure 5E

Fig. 5E should be located below Fig. 5B

-Figure 6A

Fig. 6A is not clear. Does It contain mistakes? The markers names on the top of images are probably uncorrect.

-Figure 8A

The names of protein on the left of Fig. 8A have to be re-located.

-lanes 181-182: The sentence “Notably, sample groups were found to be significantly different from the TNF-α-treated group (**p<0.01).” concerns only Fig. 4(a, b) not all Fig.4. Furthermore, the interpretation of the data is not clear enough from this sentence.

-lane 376: “FACScan” replasce with “FACS can”

-lanes 372-376: the Authors should indicate how many cells were counted for each sample.

-lanes 425-426: the english form of the sentence “This effect was associated with the way that PRFR suppressed TNF-α and induced the expression of survival, proliferation and invasive proteins.” should be modify because its signifiance is not clear enough.

6. PLOS authors have the option to publish the peer review history of their article (what does this mean?). If published, this will include your full peer review and any attached files.

Reviewer #1: No

---

## [Author Response · Author response to Decision Letter 0]

6 Nov 2019

Response to reviewer (PONE-D-19-22431)

Combination treatment of Berberine and solid lipid curcumin particles increased cell death and inhibited PI3K/Akt/mTOR pathway of human cultured glioblastoma cells more effectively than did individual treatments.

Reviewers' comments:

Reviewer's Responses to Questions

Comments to the Author

1. Is the manuscript technically sound, and do the data support the conclusions?

Reviewer #1: Partly

2. Has the statistical analysis been performed appropriately and rigorously? 

Reviewer #1: Yes

3. Have the authors made all data underlying the findings in their manuscript fully available?

The PLOS Data policy requires authors to make all data underlying the findings described in their manuscript fully available without restriction, with rare exception (please refer to the Data Availability Statement in the manuscript PDF file). The data should be provided as part of the manuscript or its supporting information or deposited to a public repository. For example, in addition to summary statistics, the data points behind means, medians and variance measures should be available. If there are restrictions on publicly sharing data—e.g. participant privacy or use of data from a third party—those must be specified.

Reviewer #1: Yes

4. Is the manuscript presented in an intelligible fashion and written in standard English?

Reviewer #1: Yes

5. Review Comments to the Author

Reviewer #1: On the whole, the work reported in the manuscript includes significant new results and represents an interesting contribution to the understanding of anti-tumor properties of natural compounds. Hence, I think that the results presented here have enough interest to be published in this Journal. However, I have a number of reservations and comments that need to be addressed before this manuscript can be accepted:

I the issues I found in this context, as it follows below (the sentences/words of the manuscript are reported in italics”: Firstly, the Authors should carefully check the text for inaccuracies and typing mistakes.

Response: We have carefully checked for typos and grammatical errors throughout the revised manuscript.

-The Authors should report the statistical test they used to verify normality and homoschedasticity of data before applying ANOVA.

Response: We have reanalyzed our data with SPSS which showed a normal distribution and homoscedasticity among the groups. Here we attached one normality and one homoscedasticity data for TUNEL staining for your review. 

TUNEL staining data for normality test

Variable 1: Vehicle; Variable 2: SLCP; Variable 3: BBR; Variable 4: BBR+SLCP

Below: Normality of Variable 2: SLCP and Variable 3: 

TUNEL staining data for Homoscedasticity test

-page 1: Title: berberine not Berberine

Response: This is corrected in the revised manuscript. Please see the revised title of the article.

-page 2: Abstract: In my opinion, the “synergistic effect” term has been used improperly and therefore should be replaced with “higher effect” or “lower effect”. Similarly, in all text synergy-derivative terms have to be corrected. In fact, a synergistic effect should be properly demonstrated and mathematically calculated (for example by using EC50). Abstract: “ascess” to be replaced with “assess”.

Response: We agree with the reviewer and changed the word “synergistic effect” to “higher effect” in the revised manuscript. Also, the typo error of the word “ascess” changed to “assess” in the revised manuscript. Please see page 2, line 13.

-page 3: Introduction: “It exhibits anticancer activity in glioma, colorectal-, lung-, prostate- and ovarian cancer, by inducing apoptotic cell death (20)” I think that other specific references should be added to (20).

Response: We have added the references of anticancer role of berberine against all the above types of cancers mentioned in the revised manuscript. Please see the page number 3, para 2 in the revised manuscript.

Introduction: “Recently, Wang and colleagues reported that BBR induces autophagy in GB by

targeting the AMPK/mTOR/ULK1-pathway (21).”

A recent paper (Agnarelli et al., 2018 Scientific Reports) showed that berberine induced autophagy in U343 GB cells. For this reason, I would suggest to cite this reference in the text, as an example: "It has been demonstrated that BBR induces autophagy in different GB cell lines U87, U251, U343 (21; Agnarelli et al., 2018). In particular, Wang and colleagues reported that BBR induces autophagy in U87 and U251 cells by targeting the AMPK/mTOR/ULK1-pathway (21).

Response: We thank the reviewer for this helpful suggestion. We have added this reference; please see reference 28 in the revised manuscript.

Introduction: “Similarly, Jin and colleagues reported that BBR inhibits angiogenesis in GB xenografts”.

I suggest to replace “Similarly” with “In addition”.

Response: This is corrected, please see page 3, para 2.

-page 4: Introduction: “Recently, we have shown that by using solid lipid curcumin particles (SLCPs), stronger anti-cancer effects and inhibition of the PI3k/Akt/mTOR pathway in human GB cell were observed than when natural Cur was used [20]” In the text the Authors wrote: "we have shown" but the reference (20) is Meeran et al., (2008).

Response: We apologize for the wrong reference. We have added our recent published work, please see the revised reference 29. 

-page 6: Materials and methods. 2.3: “whereas combination treatment of SLCP and BBR was 1:5 (20 μM + 100 μM).” The Authors should explain/comment why they choosed the combination with 1:5 ratio SLCP: BBR and analyzed the effects of this combination.

Response: We have added the description of the dose-response study we conducted for both berberine and solid lipid curcumin particles (SLCP). Based on the cell viability data we observed that in the case of SLCP after increase the concentration of SLCP above 20 µM, we did achieve more cell death. Similarly, phenomenon was observed in the case of berberine above 100 µM concentration. Therefore, we decided to use 20 µM of SLCP and 100 µM of berberine (1:5). These sentences are added in the revised manuscript, please see page 6, para 2.

-page 7: Materials and methods. 2.7: “The total number of cells and the number of annexin-V-positive cells (green) were counted per microscopic field and expressed as a percentage of dead cells.” As BBR emits fluorescent green light when appropriately excited, I suggest using Annexin V red fluorescent staining in order to analyze apoptotic cell death.

Response: We agree with reviewer, however, the green fluorescent emitted by berberine or Cur or their cotreatment was significantly lower when compared with the signal produced by Annexin-V positive cells. We have added the images after treatment with vehicle, SLCP, BBR and SLCP+BBR in Fig 4E in the revised manuscript and also added these sentences in the revised manuscript “We have used Annexin-V-FITC staining kit to detect apoptotic death which produce green fluorescent signal, whereas both BBR and SLCP treatment also might produce green fluorescent signal which could interfere with Annexin-V staining. Therefore, we imaged the cells exactly same way after treatment with BBR and SLCP. We observed a weak green signal produced by BBR and or SLCP (Fig 4E), however the green fluorescent signal produced by BBR and or SLCP was negligible in comparison to Annexin-V positive cells. Although Annexin-V with red fluorescent dye might be used to visualize apoptotic cells more efficiently than does Annexin-V-FITC after SLCP or BBR treatment”. Please see page 16, para 2 in the revised manuscript.

-page 7: Materials and methods. 2.9: In order to assess the impact of treatments on intracellular ROS level, I suggest using a different method based on measurement of the fluorescence intensity using a Multilabel Plate Reader.

Response: We agree reviewer that there are other biochemical methods available to detect ROS levels. However, using this technique, we have reported increases in ROS levels after treatment with Curcumin, previously. Please see Ref 29 and 56.

-page 9: Results 3.3: “in U-87MG cells (Fig 2A and C)” have to be replaced with “in U-87MG cells (Fig. 3A and C)”

-page 9: Results 3.3: “of their treatment (Fig 2A and C)” have to be replaced with “of their treatment (Fig. 3B and D)”

Response: These are corrected. Please see page 10, section 3.3 in the revised manuscript.

-page 10-11: Results 3.6: “We measured MMP by staining both GB cell liness with JC-1”. In my opinion, JC-1 should be used as a mitochondrial marker not as an MMP indicator (in fact the Authors did not show MMP measurements in Results). I suggest to use a biochemical method to measure a possible mitochondrial dysfunction.

Response: We agree with reviewer that biochemical method can be applied to measure MMP, however, the membrane-permeant JC-1 dye is widely used by many researchers in monitoring the mitochondrial health, especially investigating different types of cell death, including apoptosis. Therefore, JC-1 can be used as an indicator for mitochondrial health, rather than a measure of mitochondrial membrane potential. We have corrected the term “MMP” to “mitochondrial health” in the revised manuscript.

As a minor correction, “liness” have to be replaced with “lines”.

Response: This is corrected. Please see page 5, section 2.2 in the revised manuscript.

-page 11: Results 3.6: (Graph D and F) have to be corrected with (Fig. 6D and F).

Response: This is corrected. Please see revised Fig 6, please see page 12, para 2 in the revised manuscript.

-page 12: Results 3.9: The p53 pattern is composed of different size bands. In addition, the p53 expression pattern observed in the two glioblastoma cell lines is different. The Authors should comment these observations.

Response: We have corrected the p53 blots in the revised manuscript. Please see revised Fig 9

-page 13-19: Discussion, Fig. 11 and Conclusion have to be modified accordingly to suggestions.

Response: We renamed Fig 11 to Schematic diagram-1 in the revised manuscript. The discussion and conclusions are rewritten in the revised manuscript. Please see page 16 and 18

-Figure 1A: Pyknotic nuclei are not clearly visible. The authors should show higher magnified images in order to demonstrate the presence of pyknotic cells.

Response: We agree with reviewer. We produced higher magnified image for pyknotic cells. Please see revised Fig 1F. 

-Figure 4D: LDH-related results are not described in the text.

Response: We apologize for our mistake. We have added a description about the finding of our LDH release assay in page 11, para 1, line 11.

-Figure 5E: Fig. 5E should be located below Fig. 5B

Response: We rearranged Fig 5. Please see revised Fig 5.

-Figure 6A: Fig. 6A is not clear. Does It contain mistakes? The markers names on the top of images are probably incorrect.

Response: Yes, it was our mistake. We corrected this figure in the revised manuscript. Please see revised Fig 6A.

-Figure 8A: The names of protein on the left of Fig. 8A have to be re-located.

Response: We rearranged the Fig 8A. Please see revised Fig 8A.

---

## [Decision Letter · Decision Letter 1]

11 Nov 2019

Combination treatment of berberine and solid lipid curcumin particles increased cell death and inhibited PI3K/Akt/mTOR pathway of human cultured glioblastoma cells more effectively than did individual treatments

PONE-D-19-22431R1

Dear Dr. Maiti,

We are pleased to inform you that your manuscript has been judged scientifically suitable for publication and will be formally accepted for publication once it complies with all outstanding technical requirements.

With kind regards,

Ferenc Gallyas, Jr., Ph.D., D.Sc.

Section Editor

PLOS ONE

Additional Editor Comments (optional):

Reviewers' comments:

Reviewer's Responses to Questions

**Comments to the Author**

1. If the authors have adequately addressed your comments raised in a previous round of review and you feel that this manuscript is now acceptable for publication, you may indicate that here to bypass the “Comments to the Author” section, enter your conflict of interest statement in the “Confidential to Editor” section, and submit your "Accept" recommendation.

Reviewer #1: All comments have been addressed

2. Is the manuscript technically sound, and do the data support the conclusions?

Reviewer #1: Yes

3. Has the statistical analysis been performed appropriately and rigorously? 

Reviewer #1: Yes

4. Have the authors made all data underlying the findings in their manuscript fully available?

Reviewer #1: Yes

5. Is the manuscript presented in an intelligible fashion and written in standard English?

Reviewer #1: Yes

6. Review Comments to the Author

Reviewer #1: The authors addressed my suggestions. I believe that this manuscript is acceptable for publication in PLOS ONE Journal.

7. PLOS authors have the option to publish the peer review history of their article (what does this mean?). If published, this will include your full peer review and any attached files.

Reviewer #1: No

---

## [Editor Report · Acceptance letter]

27 Nov 2019

PONE-D-19-22431R1 

Combination treatment of berberine and solid lipid curcumin particles increased cell death and inhibited PI3K/Akt/mTOR pathway of human cultured glioblastoma cells more effectively than did individual treatments 

Dear Dr. Maiti:

I am pleased to inform you that your manuscript has been deemed suitable for publication in PLOS ONE. Congratulations! Your manuscript is now with our production department. 

With kind regards,

on behalf of

Dr. Ferenc Gallyas, Jr. 

Section Editor

PLOS ONE